# TRANSFORMER MEETS BOUNDARY VALUE INVERSE PROBLEMS

**Ruchi Guo**
Department of Mathematics
University of California, Irvine

**Shuhao Cao**
Division of Computing, Analytics, and Mathematics
School of Science and Engineering
University of Missouri-Kansas City

**Long Chen**
Department of Mathematics
University of California, Irvine

## ABSTRACT

A Transformer-based deep direct sampling method is proposed for electrical impedance tomography, a well-known severely ill-posed nonlinear boundary value inverse problem. A real-time reconstruction is achieved by evaluating the learned inverse operator between carefully designed data and the reconstructed images. An effort is made to give a specific example to a fundamental question: whether and how one can benefit from the theoretical structure of a mathematical problem to develop task-oriented and structure-conforming deep neural networks? Specifically, inspired by direct sampling methods for inverse problems, the 1D boundary data in different frequencies are preprocessed by a partial differential equation-based feature map to yield 2D harmonic extensions as different input channels. Then, by introducing learnable non-local kernels, the direct sampling is recast to a modified attention mechanism. The new method achieves superior accuracy over its predecessors and contemporary operator learners and shows robustness to noises in benchmarks. This research shall strengthen the insights that, despite being invented for natural language processing tasks, the attention mechanism offers great flexibility to be modified in conformity with the a priori mathematical knowledge, which ultimately leads to the design of more physics-compatible neural architectures.

## 1 INTRODUCTION

Boundary value inverse problems aim to recover the internal structure or distribution of multiple media inside an object (2D reconstruction) based on only the data available on the boundary (1D signal input), which arise from many imaging techniques, e.g., electrical impedance tomography (EIT) (Holder, 2004), diffuse optical tomography (DOT) (Culver et al., 2003), magnetic induction tomography (MIT) (Griffiths et al., 1999). Not needing any internal data renders these techniques generally non-invasive, safe, cheap, and thus quite suitable for monitoring applications.

In this work, we shall take EIT as an example to illustrate how a more structure-conforming neural network architecture leads to better results in certain physics-based tasks. Given a 2D bounded domain $\Omega$ and an inclusion $D$, the forward model is the following partial differential equation (PDE)

$$\nabla \cdot (\sigma \nabla u) = 0 \quad \text{in } \Omega, \quad \text{where } \sigma = \sigma_1 \text{ in } D, \text{ and } \sigma = \sigma_0 \text{ in } \Omega \backslash \overline{D}, \tag{1}$$

where $\sigma$ is a piecewise constant function defined on $\Omega$ with known function values $\sigma_0$ and $\sigma_1$, but the shape of the inclusion $D$ buried in $\Omega$ is unknown. The goal is to recover the shape of $D$ using only the boundary data on $\partial\Omega$ (Figure 1). Specifically, by exerting a current $g$ on the boundary, one solves (1) with the Neumann boundary condition $\sigma\nabla u \cdot \mathbf{n}|_{\partial\Omega} = g$, where $\mathbf{n}$ is the outwards unit normal direction of $\partial\Omega$, to get a unique $u$ on the whole domain $\Omega$. In practice, only the Dirichlet boundary value representing the voltages $f = u|_{\partial\Omega}$ on the boundary can be measured. This procedure is called Neumann-to-Dirichlet (NtD) mapping:

$$\Lambda_\sigma : H^{-1/2}(\partial\Omega) \to H^{1/2}(\partial\Omega), \quad \text{with } g = \sigma\nabla u \cdot \mathbf{n}|_{\partial\Omega} \mapsto f = u|_{\partial\Omega}. \tag{2}$$

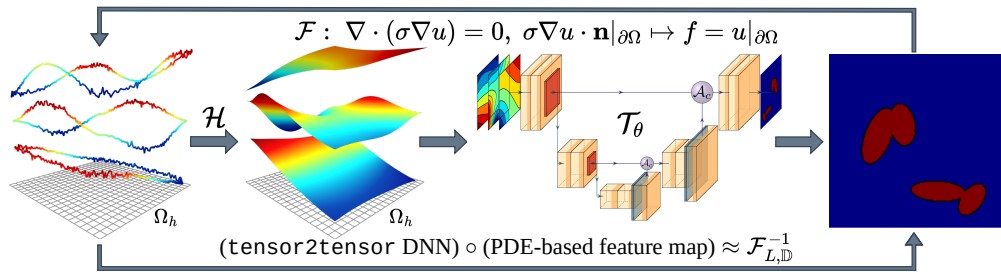

Figure 1: Schematics of the pipeline: the approximation of the inverse operator is decomposed into (i) a PDE-based feature map $\mathcal{H}$: the 1D boundary data is extended to 2D features by a harmonic extension; (ii) a tensor2tensor neural network $\mathcal{T}_\theta$ that outputs the reconstruction.

For various notation and the Sobolev space formalism, we refer readers to Appendix A; for a brief review of the theoretical background of EIT we refer readers to Appendix B. The NtD map above in (2) can be expressed as

$$\mathbf{f} = \mathbf{A}_\sigma \mathbf{g}, \qquad (3)$$

where $\mathbf{g}$ and $\mathbf{f}$ are (infinite-dimensional) vector representations of functions $g$ and $f$ relative to a chosen basis, and $\mathbf{A}_\sigma$ is the matrix representation of $\Lambda_\sigma$ (see Appendix B for an example).

The original mathematical setup of EIT is to use the NtD map $\Lambda_\sigma$ in (2) to recover $\sigma$, referred to as the case of full measurement (Calderón, 2006). In this case, the forward and inverse operators associated with EIT can be formulated as

$$\mathcal{F} \,:\, \sigma \mapsto \Lambda_\sigma, \quad \text{and} \quad \mathcal{F}^{-1} \,:\, \Lambda_\sigma \mapsto \sigma. \qquad (4)$$

Fix a set of basis $\{g_l\}_{l=1}^\infty$ of the corresponding Hilbert space containing all admissible currents. Then, mathematically speaking, "knowing the operator $\Lambda_\sigma$" means that one can measure all the current-to-voltage pairs $\{g_l, f_l := \Lambda_\sigma g_l\}_{l=1}^\infty$ and construct the infinite-dimensional matrix $\mathbf{A}_\sigma$.

However, as infinitely many boundary data pairs are not attainable in practice, the problem of more practical interest is to use only a few data pairs $\{(g_l, f_l)\}_{l=1}^L$ for reconstruction. In this case, the forward and inverse problems can be formulated as

$$\mathcal{F}_L : \sigma \mapsto \{(g_1, \Lambda_\sigma g_1), ..., (g_L, \Lambda_\sigma g_L)\} \quad \text{and} \quad \mathcal{F}_L^{-1} : \{(g_1, \Lambda_\sigma g_1), ..., (g_L, \Lambda_\sigma g_L)\} \mapsto \sigma. \qquad (5)$$

For limited data pairs, the inverse operator $\mathcal{F}_L^{-1}$ is extremely ill-posed or even not well-defined (Isakov & Powell, 1990; Barceló et al., 1994; Kang & Seo, 2001; Lionheart, 2004); namely, the same boundary measurements may correspond to different $\sigma$. In view of the matrix representation $\mathbf{A}_\sigma$, for $\mathbf{g}_l = \mathbf{e}_l, l = 1, ..., L$, with $\mathbf{e}_l$ being unit vectors of a chosen basis, $(\mathbf{f}_1, ..., \mathbf{f}_L)$ only gives the first $L$ columns of $\mathbf{A}_\sigma$. It is possible that two matrices $\mathbf{A}_\sigma$ and $\mathbf{A}_{\tilde{\sigma}}$ have similar first $L$ columns but $\|\sigma - \tilde{\sigma}\|$ is large.

How to deal with this "ill-posedness" is a central theme in boundary value inverse problem theories. The operator learning approach has the potential to tame the ill-posedness by restricting $\mathcal{F}_L^{-1}$ at a set of sampled data $\mathbb{D} := \{\sigma^{(k)}\}_{k=1}^N$, with different shapes and locations following certain distribution. Then the problem becomes to approximate

$$\mathcal{F}_{L,\mathbb{D}}^{-1} : \{(g_1, \Lambda_{\sigma^{(k)}} g_1), ..., (g_L, \Lambda_{\sigma^{(k)}} g_L)\} \mapsto \sigma^{(k)}, \quad k = 1, \ldots, N. \qquad (6)$$

The fundamental assumption here is that this map is "well-defined" enough to be regarded as a high-dimensional interpolation (learning) problem on a compact data submanifold (Seo et al., 2019; Ghattas & Willcox, 2021), and the learned approximate mapping can be evaluated at newly incoming $\sigma$'s. The incomplete information of $\Lambda_\sigma$ due to a small $L$ for one single $\sigma$ is compensated by a large $N \gg 1$ sampling of different $\sigma$'s.

## 2 BACKGROUND, RELATED WORK, AND CONTRIBUTIONS

**Classical iterative methods.** There are in general two types of methodology to solve inverse problems. The first one is a large family of iterative or optimization-based methods (Dobson &

Santosa, 1994; Martin & Idier, 1997; Chan & Tai, 2003; Vauhkonen et al., 1999; Guo et al., 2019; Rondi & Santosa, 2001; Chen et al., 2020; Bao et al., 2020; Gu et al., 2021). One usually looks for an approximated $\sigma$ by solving a minimization problem with a regularization $\mathcal{R}(\sigma)$ to alleviate the ill-posedness, say

$$\inf_\sigma \left\{ \sum_{l=1}^{L} \|\Lambda_\sigma g_l - f_l\|_{\partial\Omega}^2 + \mathcal{R}(\sigma) \right\}. \tag{7}$$

The design of regularization $\mathcal{R}(\sigma)$ plays a critical role in a successful reconstruction (Tarvainen et al., 2008; Tehrani et al., 2012; Wang et al., 2012). Due to the ill-posedness, the computation for almost all iterative methods usually takes numerous iterations to converge, and the reconstruction is highly sensitive to noise. Besides, the forward operator $\mathcal{F}(\cdot)$ needs to be evaluated at each iteration, which is itself expensive as it requires solving forward PDE models.

**Classical direct methods.** The second methodology is to develop a well-defined mapping $\mathcal{G}_\theta$ parametrized by $\theta$, empirically constructed to approximate the inverse map itself, say $\mathcal{G}_\theta \approx \mathcal{F}^{-1}$. These methods are referred to as non-iterative or direct methods in the literature. Distinguished from iterative approaches, direct methods are typically highly problem-specific, as they are designed based on specific mathematical structures of their respective inverse operators. For instance, methods in EIT and DOT include factorization methods (Kirsch & Grinberg, 2007; Azzouz et al., 2007; Brühl, 2001; Hanke & Brühl, 2003), MUSIC-type algorithms (Cheney, 2001; Ammari & Kang, 2004; 2007; Lee et al., 2011), and the D-bar methods (Knudsen et al., 2007; 2009) based on a Fredholm integral equation (Nachman, 1996), among which are the direct sampling methods (DSM) being our focus in this work (Chow et al., 2014; 2015; Kang et al., 2018; Ito et al., 2013; Ji et al., 2019; Harris & Kleefeld, 2019; Ahn et al., 2020; Chow et al., 2021; Harris et al., 2022). These methods generally have a closed-form $\mathcal{G}_\theta$ for approximation, and the parameters $\theta$ represent model-specific mathematical objects. For each fixed $\theta$, this procedure is usually much more stable than iterative approaches with respect to the input data. Furthermore, the evaluation for each boundary data pair is distinctly fast, as no optimization is needed. However, a simple closed-form $\mathcal{G}_\theta$ admitting efficient execution may not be available in practice since some mathematical assumptions and derivation may not hold. For instance, MUSIC-type and D-bar methods generally require an accurate approximation to $\Lambda_\sigma$, while DSM poses restrictions on the boundary data, domain geometry, etc., see Appendix D for details.

**Boundary value inverse problems.** For most cases of boundary value inverse problems in 2D, the major difference, e.g., with an inverse problem in computer vision (Marroquin et al., 1987), is that data are only available on 1D manifolds, which are used to reconstruct 2D targets. When comparing (4) with a linear inverse problem in signal processing $\mathbf{y} = A\mathbf{x} + \boldsymbol{\epsilon}$, to recover a signal $\mathbf{x}$ from measurement $\mathbf{y}$ with noise $\boldsymbol{\epsilon}$, the difference is more fundamental in that $\mathcal{F}(\cdot)$ itself is highly nonlinear and involves boundary value PDEs. Moreover, the boundary data themselves generally involve certain input-output structures (NtD maps), which adds more complexity. In Adler & Guardo (1994); Fernández-Fuentes et al. (2018); Feng et al. (2018), boundary measurements are collected and directly input into feedforward fully connected networks. As the data reside on different manifolds, special treatments are made to the input data, such as employing pre-reconstruction stages to generate rough 2D input to CNNs (Ben Yedder et al., 2018; Ren et al., 2020; Pakravan et al., 2021).

**Deep neural network and inverse problems.** Solving an inverse problem is essentially to give a satisfactory approximation to $\mathcal{F}^{-1}$ but based on finitely many measurements. The emerging deep learning (DL) based on Deep Neural Networks (DNN) to directly emulate operators significantly resembles those classical direct methods mentioned above. However, operator learners by DNNs are commonly considered black boxes. A natural question is how the a priori mathematical knowledge can be exploited to design more physics-compatible DNN architectures. In pursuing the answer to this question, we aim to provide a supportive example that bridges deep learning techniques and classical direct methods, which improves the reconstruction of EIT.

**Operator learners.** Operator learning has become an active research field for inverse problems in recent years, especially related to image reconstruction where CNN plays a central role, see, e.g., Kłosowski & Rymarczyk (2017); Nguyen et al. (2018); Tan et al. (2018); Jin et al. (2017); Kang et al. (2017); Barbastathis et al. (2019); Latif et al. (2019); Zhu et al. (2018); Chen et al. (2021); Coxson et al. (2022); Zhu et al. (2023b). Notable examples of efforts to couple classical reconstruction methods and CNN include Hamilton et al. (2019); Hamilton & Hauptmann (2018), where a CNN post-processes images obtained by the classical D-bar methods, and Fan et al. (2019); Fan & Ying (2020) where BCR-Net is developed to mimic pseudo-differential operators appearing in many inverse

problems. A deep direct sampling method is proposed in Guo & Jiang (2020); Guo et al. (2021) that learns local convolutional kernels mimicking the gradient operator of DSM. Another example is radial basis function neural networks seen in Hrabuska et al. (2018); Michalikova et al. (2014); Wang et al. (2021a). Nevertheless, convolutions in CNNs use kernels whose receptive fields involve only a small neighborhood of a pixel. Thus, layer-wise, CNN does not align well with the non-local nature of inverse problems. More recently, the learning of PDE-related forward problems using global kernel has gained attraction, most notably the Fourier Neural Operator (FNO) (Nelsen & Stuart, 2021; Li et al., 2021a; Kovachki et al., 2021; Guibas et al., 2022; Zhao et al., 2022; Wen et al., 2022; Li et al., 2022b). FNO takes advantage of the low-rank nature of certain problems and learns a local kernel in the frequency domain yet global in the spatial-temporal domain, mimicking the solution's kernel integral form. Concurrent studies include DeepONets (Lu et al., 2021; Wang et al., 2021b; Jin et al., 2022b), Transformers (Cao, 2021; Kissas et al., 2022; Li et al., 2022a; Liu et al., 2022; Fonseca et al., 2023), Integral Autoencoder (Ong et al., 2022), Multiwavelet Neural Operators (Gupta et al., 2021; 2022), and others (Lütjens et al., 2022; Hu et al., 2022; Boussif et al., 2022; de Hoop et al., 2022a;b; Ryck & Mishra, 2022; Seidman et al., 2022; Zhang et al., 2023; Lee, 2023; Zhu et al., 2023a).

**Related studies on Transformers.** The attention mechanism-based models have become state of the art in many areas since Vaswani et al. (2017). One of the most important and attractive aspects of the attention mechanism is its unparalleled capability to efficiently model non-local long-range interactions (Katharopoulos et al., 2020; Choromanski et al., 2021; Nguyen et al., 2021). The relation of the attention with kernel learning is first studied in Tsai et al. (2019) and later connected with random feature (Peng et al., 2021). Connecting the non-PDE-based integral kernels and the attention mechanism has been seen in Hutchinson et al. (2021); Guibas et al. (2022); Nguyen et al. (2022); Han et al. (2022). Among inverse problems, Transformers have been applied in medical imaging applications, including segmentation (Zhou et al., 2021; Hatamizadeh et al., 2022; Petit et al., 2021), X-Ray (Tanzi et al., 2022), magnetic resonance imaging (MRI) (He et al., 2022), ultrasound (Perera et al., 2021), optical coherence tomography (OCT) (Song et al., 2021). To our best knowledge, no work in the literature establishes an architectural connection between the attention mechanism in Transformer and the mathematical structure of PDE-based inverse problems.

## 2.1 CONTRIBUTIONS

- *A structure-conforming network architecture.* Inspired by the EIT theory and classic DSM, we decompose the approximation of the inverse operator into a harmonic extension and an integral operator with learnable non-local kernels that has an attention-like structure. Additionally, the attention architecture is reinterpreted through a Fredholm integral operator to rationalize the application of the Transformer to the boundary value inverse problem.

- *Theoretical and experimental justification for the advantage of Transformer.* We have proved that, in Transformers, modified attention can represent target functions exhibiting higher frequency natures from lower frequency input features. A comparative study in the experiments demonstrates a favorable match between the Transformer and the benchmark problem.

## 3 INTERPLAY BETWEEN MATHEMATICS AND NEURAL ARCHITECTURES

In this section, we try to articulate that the triple tensor product in the attention mechanism matches exceptionally well with representing a solution in the inverse operator theory of EIT. In pursuing this end goal, this study tries to answer the following motivating questions:

(**Q1**) What is an appropriate finite-dimensional data format as inputs to the neural network?

(**Q2**) Is there a suitable neural network matching the mathematical structure?

### 3.1 FROM EIT TO OPERATOR LEARNING

In the case of full measurement, the operator $\mathcal{F}^{-1}$ can be well approximated through a large number of $(\sigma, \mathbf{A}_\sigma)$ data pairs. This mechanism essentially results in a *tensor2tensor* mapping/operator from $\mathbf{A}_\sigma$ to the imagery data representing $\sigma$. In particular, the BCR-Net (Fan & Ying, 2020) is a DNN approximation falling into this category. However, when there are very limited boundary data pairs accessible, the task of learning the full matrix $\mathbf{A}_\sigma$ becomes obscure, which complicates the development of a *tensor2tensor* pipeline for operator learning.

**Operator learning problems for EIT.** We first introduce several attainable approximations of infinite-dimensional spaces by finite-dimensional counterparts for the proposed method.

(1) *Spatial discretization.* Let $\Omega_h$ be a mesh of $\Omega$ with the mesh spacing $h$ and let $\{z_j\}_{j=1}^M := \mathcal{M}$ be the set of grid points to represent the 2D discretization of continuous signals. Then a function $u$ defined almost everywhere in $\Omega$ can be approximated by a vector $\mathbf{u}_h \in \mathbb{R}^M$.

(2) *Sampling of D.* We generate $N$ samples of $D$ with different shapes and locations following certain distributions. For example, elliptical inclusions with random semi-axes and centers are generated as a benchmark (see Appendix C.1 for details). With the known $\sigma_0$ and $\sigma_1$, set the corresponding data set $\mathbb{D} = \{\sigma^{(1)}, \sigma^{(2)}, ..., \sigma^{(N)}\}$. $N$ is usually large enough to represent field applications of interest.

(3) *Sampling of NtD maps.* For the $k$-th sample of $D$, we generate $L$ pairs of boundary data $\{(g_l^{(k)}, f_l^{(k)})\}_{l=1}^L$ by solving PDE (1), which can be thought of as sampling of columns of the infinite matrix $\mathbf{A}_\sigma$ representing the NtD map. By the proposed method, $L$ can be chosen to be very small ($\leq 3$) to yield satisfactory results.

Our task is to find a parameterized mapping $\mathcal{G}_\theta$ to approximate $\mathcal{F}_{L,\mathbb{D}}^{-1}$ (6) by minimizing

$$\mathcal{J}(\theta) := \frac{1}{N} \sum_{k=1}^N \|\mathcal{G}_\theta(\{(g_l^{(k)}, f_l^{(k)})\}_{l=1}^L) - \sigma^{(k)}\|^2, \tag{8}$$

for a suitable norm $\|\cdot\|$. Hyper-parameters $N, h, L$ will affect the finite-dimensional approximation to the infiniteness in the following way: $h$ determines the resolution to approximate $D$; $N$ affects the representativity of the training data set; $L$ decides how much of a finite portion of the infinite spectral information of $\Lambda_\sigma$ can be accessed.

### 3.2 FROM HARMONIC EXTENSION TO TENSOR-TO-TENSOR

To establish the connection between the problem of interest with the attention used in the Transformers, we first construct higher-dimensional tensors from the 1D boundary data. The key is a harmonic extension of the boundary data that can be viewed as a PDE-based feature map. We begin with a theorem to motivate it.

Let $\mathcal{I}^D$ be the characteristic function of $D$ named *index function*, i.e., $\mathcal{I}^D(x) = 1$ if $x \in D$ and $\mathcal{I}^D(x) = 0$ if $x \notin D$. Thus, $\sigma$ can be directly identified by the shape of $D$ through the formula $\sigma = \sigma_1 \mathcal{I}^D + \sigma_0(1 - \mathcal{I}^D)$. In this setup, reconstructing $\sigma$ is equivalent to reconstructing $\mathcal{I}^D$.

Without loss of generality, we let $\sigma_1 > \sigma_0$. $\Lambda_{\sigma_0}$ is understood as the NtD map with $\sigma = \sigma_0$ on the whole domain, i.e., it is taken as the known background conductivity (no inclusion), and thus $\Lambda_{\sigma_0} g$ can be readily computed. Then $f - \Lambda_{\sigma_0} g = (\Lambda_\sigma - \Lambda_{\sigma_0})g$ measures the difference between the NtD mappings and encodes the information of $\sigma$. The operator $\Lambda_\sigma - \Lambda_{\sigma_0}$ is positive definite, and it has eigenvalues $\{\lambda_l\}_{l=1}^\infty$ with $\lambda_1 > \lambda_2 > \cdots > 0$ (Cheng et al., 1989).

**Theorem 1.** *Suppose that the 1D boundary data $g_l$ is the eigenfunction of $\Lambda_\sigma - \Lambda_{\sigma_0}$ corresponding to the $l$-th eigenvalue $\lambda_l$, and let the 2D data functions $\phi_l$ be obtained by solving*

$$-\Delta\phi_l = 0 \quad in \quad \Omega, \quad \mathbf{n} \cdot \nabla\phi_l = (f_l - \Lambda_{\sigma_0} g_l) \quad on \quad \partial\Omega, \quad \int_{\partial\Omega} \phi_l \, ds = 0, \tag{9}$$

*for $l = 1, 2, \ldots$. Define the space $\widetilde{\mathbb{S}}_L = \mathrm{Span}\{\partial_{x_1}\phi_l \, \partial_{x_2}\phi_l : l = 1, \ldots, L\}$, and the dictionary $\mathbb{S}_L = \{a_1 + a_2 \arctan(a_3 v) : v \in \widetilde{\mathbb{S}}_L, a_1, a_2, a_3 \in \mathbb{R}\}$. Then, for any $\epsilon > 0$, we con construct an index function $\mathcal{I}_L^D \in \mathbb{S}_L$ s.t. $\sup_{x \in \Omega} |\mathcal{I}^D(x) - \mathcal{I}_L^D(x)| \leq \epsilon$ provided $L$ is large enough.*

The full proof of Theorem 1 can be found in Appendix E. This theorem gives a constructive approach for approximating $\mathcal{F}^{-1}$ and justifies the practice of approximating $\mathcal{I}^D$ when $L$ is large enough. The function $\phi_l$ is called the *harmonic extension* of $f_l - \Lambda_{\sigma_0} g_l$.

On the other hand, it relies on "knowing" the entire NtD map $\Lambda_\sigma$ to construct $\mathcal{I}_L^D$ explicitly. Namely, the coefficients of $\partial_{x_1}\phi_l \, \partial_{x_2}\phi_l$ depend on a big chunk of spectral information (eigenvalues and eigenfunctions) of $\Lambda_\sigma$, which may not be available in practice. Thus, the mathematics itself in this theorem does not provide an architectural hint on building a structure-conforming DNN.

To further dig out the hidden structure, we focus on the case of a single measurement, i.e., $L = 1$. With this setting, it is possible to derive an explicit and simple formula to approximate $\mathcal{I}^D$ which

is achieved by the classical direct sampling methods (DSM) (Chow et al., 2014; 2015; Kang et al., 2018; Ito et al., 2013; Ji et al., 2019; Harris & Kleefeld, 2019; Ahn et al., 2020). For EIT,

$$\mathcal{I}^D(x) \approx \mathcal{I}_1^D(x) := R(x)\left(\mathbf{d}(x) \cdot \nabla\phi(x)\right) \quad x \in \Omega, \, \mathbf{d}(x) \in \mathbb{R}^2, \tag{10}$$

is derived in Chow et al. (2014), where (see a much more detailed formulation in Appendix D)

- $\phi$ is the harmonic extension of $f - \Lambda_{\sigma_0} g$ with certain noise $f - \Lambda_{\sigma_0} g + \xi$;
- $\mathbf{d}(x)$ is called a probing direction and can be chosen empirically as $\mathbf{d}(x) = \nabla\phi(x)/\|\nabla\phi(x)\|$;
- $R(x) = (\|f - \Lambda_{\sigma_0}g\|_{\partial\Omega}|\eta_x|_Y)^{-1}$, where $\eta_x$ is a function of $\mathbf{d}(x)$ and measured in $|\cdot|_Y$ semi-norm on boundary $\partial\Omega$.

Both $\phi$ and $\eta_x$ can be computed effectively by traditional fast PDE solvers, such as finite difference or finite element methods based on $\Omega_h$ in Section 3.1. However, the reconstruction accuracy is much limited by a single measurement, the nonparametric ansatz, and empirical choices of $\mathbf{d}(x)$ and $|\cdot|_Y$. These restrictions leave room for DL methodology. See Appendix D for a detailed discussion.

Constructing harmonic extension (2D features) from boundary data (1D signal input with limited depth) can contribute to the desired high-quality reconstruction. First, harmonic functions are highly smooth away from the boundary, of which the solution automatically smooths out the noise on the boundary due to PDE theory (Gilbarg & Trudinger, 2001, Chapter 8), and thus make the reconstruction highly robust with respect to the noise (e.g., see Figure 3 in Appendix C.1). Second, in terms of using certain backbone networks to generate features for downstream tasks, harmonic extensions can be understood as a problem-specific way to design higher dimensional feature maps (Álvarez et al., 2012), which renders samples more separable in a higher dimensional data manifold than the one with merely boundary data. See Figure 1 to illustrate this procedure.

The information of $\sigma$ is deeply hidden in $\phi$. As shown in Figure 1 (see also Appendix C), one cannot observe any pattern of $\sigma$ directly from $\phi$. It is different from and more challenging than the inverse problems studied in (Bhattacharya et al., 2021; Khoo et al., 2021) that aim to reconstruct 2D targets from the much more informative 2D internal data of $u$.

In summary, both Theorem 1 and the formula of DSM (10) offer inspiration to give a potential answer to (Q1): the *harmonic extension $\phi$* (2D features) of $f - \Lambda_{\sigma_0}g$ (1D measurements) naturally encodes the information of the true characteristic function $\mathcal{I}_D$ (2D targets). As there is a pointwise correspondence between the harmonic extensions and the targets at 2D grids, a tensor representation of $\nabla\phi_l$ at these grid points can then be used as the input to a *tensor2tensor*-type DNN to learn $\mathcal{I}_D$. Naturally, the grids are set as the positional embedding explicitly. In comparison, the positional information is buried more deeply in 1D measurements. As shown in Figure 1, $\mathcal{G}_\theta$ can be nicely decoupled into a composition of a learnable neural network operator $\mathcal{T}_\theta$ and a non-learnable PDE-based feature map $\mathcal{H}$, i.e., $\mathcal{G}_\theta = \mathcal{T}_\theta \circ \mathcal{H}$. The architecture of $\mathcal{T}_\theta$ shall be our interest henceforth.

### 3.3 FROM CHANNELS IN ATTENTION TO BASIS IN INTEGRAL TRANSFORM

In this subsection, a modified attention mechanism is proposed as the basic block in the tensor2tensor-type mapping introduced in the next two subsections. Its reformulation conforms with one of the most used tools in applied mathematics: the integral transform. In many applications such as inverse problems, the interaction (kernel) does not have any explicit form, which meshes well with DL methodology philosophically. In fact, this is precisely the situation of the EIT problem considered.

Let the input of an encoder attention block be $\mathbf{x}_h \in \mathbb{R}^{M \times c}$ with $c$ channels, then the query $Q$, key $K$, value $V$ are generated by three learnable projection matrices $\theta := \{W^Q, W^K, W^V\} \subset \mathbb{R}^{c \times c}$: $\diamond = \mathbf{x}_h W^\diamond, \diamond \in \{Q, K, V\}$. Here $c \gg L$ is the number of expanded channels for the latent representations. A modified dot-product attention is proposed as follows:

$$U = \text{Attn}(\mathbf{x}_h) := \alpha\left(\text{nl}_Q(Q)\text{nl}_K(K)^\top\right)V = \alpha(\widetilde{Q}\widetilde{K}^\top)V \in \mathbb{R}^{M \times c}, \tag{11}$$

where $\text{nl}_Q(\cdot)$ and $\text{nl}_K(\cdot)$ are two learnable normalizations. Different from Nguyen & Salazar (2019); Xiong et al. (2020), this pre-inner-product normalization is applied right before the matrix multiplication of query and key. This practice takes inspiration from the normalization in the index function kernel integral (10) and (20), see also Boyd (2001) where the normalization for orthogonal bases essentially uses the (pseudo)inverse of the Gram matrices. In practice, layer normalization (Ba

et al., 2016) or batch normalization (Ioffe & Szegedy, 2015) is used as a cheap alternative. Constant $\alpha = h^2$ is a mesh-based weight such that the summation becomes an approximation to an integral.

To elaborate these rationales, the $j$-th column of the $i$-th row $U_i$ of $U$ is $(U_i)^j = \alpha A_{i\bullet} \cdot V^j$, in which the $i$-th row $A_{i\bullet} = (\widetilde{Q}\widetilde{K}^\top)_{i\bullet}$ and the $j$-th column $V^j := V_{\bullet j}$. Thus, applying this to every column $1 \le j \le c$, attention (11) becomes a basis expansion representation for the $i$-th row $U_i$

$$U_i = \alpha A_{i\bullet} \begin{pmatrix} \underline{\quad} & V_1 & \underline{\quad} \\ \underline{\quad} & \vdots & \underline{\quad} \\ \underline{\quad} & V_M & \underline{\quad} \end{pmatrix} = \sum_{m=1}^{M} \alpha A_{im} V_m =: \sum_{m=1}^{M} \mathcal{A}(Q_i, K_m) V_m. \quad (12)$$

Here, $\alpha A_{i\bullet}$ contains the coefficients for the linear combination of $\{V_m\}_{m=1}^M$. This set $\{V_m\}_{m=1}^M$ forms the $V$'s row space, and it further forms each row of the output $U$ by multiplying with $A$. $\mathcal{A}(\cdot, \cdot)$ in (12) stands for the attention kernel, which aggregates the pixel-wise feature maps to measure how the projected latent representations interact. Moreover, the latent representation $U_i$ in an encoder layer is spanned by the row space of $V$ and is being nonlinearly updated cross-layer-wise.

For $\mathbf{x}_h, U, Q, K, V$, a set of feature maps are assumed to exist: for example $u(\cdot)$ maps $\mathbb{R}^2 \to \mathbb{R}^{1 \times c}$, i.e., $U_i = u(z_i) = [u_1(z_i), \cdots, u_c(z_i)]$, e.g., see Choromanski et al. (2021), then an instance-dependent kernel $\kappa_\theta(\cdot, \cdot) : \mathbb{R}^2 \times \mathbb{R}^2 \to \mathbb{R}$ can be defined by

$$\mathcal{A}(Q_i, K_j) := \alpha \langle \widetilde{Q}_i, \widetilde{K}_j \rangle = \alpha \langle q(z_i), k(z_j) \rangle =: \alpha \kappa_\theta(z_i, z_j). \quad (13)$$

Now the discrete kernel $\mathcal{A}(\cdot, \cdot)$ with tensorial input is rewritten to this kernel $\kappa_\theta(\cdot, \cdot)$, thus the dot-product attention is expressed as a nonlinear integral transform for the $l$-th channel:

$$u_l(z) = \alpha \sum_{x \in \mathcal{M}} \left( q(z) \cdot k(x) \right) v_l(x) \, \delta_x \approx \int_\Omega \kappa_\theta(z, x) v_l(x) \, d\mu(x), \quad 1 \le l \le c. \quad (14)$$

Through certain minimization such as (8), the backpropagation updates $\theta$, which further leads a new set of latent representations. This procedure can be viewed as an iterative method to update the basis residing in each channel by solving the Fredholm integral equation of the first kind in (14).

To connect attention with inverse problems, the multiplicative structure in a kernel integral form for attention (14) is particularly useful. (14) is a type of Pincherle-Goursat (degenerate) kernels (Kress, 1999, Chapter 11) and approximates the full kernel using only a finite number of bases. The number of learned basis functions in expansion (12) depends on the number of channels $n$. Here we show the following theorem; heuristically, it says that: given enough but finite channels of latent representations, the attention kernel integral can "bootstrap" in the frequency domain, that is, generating an output representation with higher frequencies than the input. Similar approximation results are impossible for layer-wise propagation in CNN if one opts for the usual framelet/wavelet interpretation (Ye et al., 2018). For example, if there are no edge-like local features in the input (see for empirical evidence in Figure 9 and Figure 10), a single layer of CNN filters without nonlinearity cannot learn weights to extract edges. The full proof with a more rigorous setting is in Appendix F.

**Theorem 2** (Frequency bootstrapping). *Suppose there exists a channel $l$ in $V$ such that $(V_i)_l = \sin(a z_i)$ for some $a \in \mathbb{Z}^+$, the current finite-channel sum kernel $\mathcal{A}(\cdot, \cdot)$ approximates a non-separable kernel to an error of $O(\epsilon)$ under certain norm $\|\cdot\|_X$. Then, there exists a set of weights such that certain channel $k'$ in the output of (12) approximates $\sin(a'z)$, $\mathbb{Z}^+ \ni a' > a$ with an error of $O(\epsilon)$ under the same norm.*

The considered inverse problem is essentially to recover higher-frequency eigenpairs of $\Lambda_\sigma$ based on lower-frequency data, see, e.g., Figure 1. $\Lambda_\sigma$ together with all its spectral information can be determined by the recovered inclusion shape. Thus, the existence in Theorem 2 partially justifies the advantages of adopting the attention mechanism for the considered problem.

### 3.4 FROM INDEX FUNCTION INTEGRAL TO TRANSFORMER

In (10), the probing direction $\mathbf{d}(x)$, the inner product $\mathbf{d}(x) \cdot \nabla\phi(x)$, and the norm $|\cdot|_Y$ are used as ingredients to form certain non-local instance-based learnable kernel integration. This non-localness is a fundamental trait for many inverse problems, in that $\mathcal{I}^D(x)$ depends on the entire data function. Then, the discretization of the modified index function is shown to match the multiplicative structure of the modified attention mechanism in (11).

In the forthcoming derivations, $\mathcal{K}(x, y)$, $\mathcal{Q}(x, y)$, and a self-adjoint positive definite linear operator $\mathcal{V} : L^2(\partial\Omega) \to L^2(\partial\Omega)$, are shown to yield the emblematic Q-K-V structure of attention. To this end, we make the following modifications and assumptions to the original index function in (10).

- The reformulation of the index function is motivated by the heuristics that the agglomerated global information of $\phi$ could be used as "keys" to locate a point $x$.

$$\hat{\mathcal{I}}_1^D(x) := R(x) \int_\Omega \mathbf{d}(x) \cdot \mathcal{K}(x,y) \nabla \phi(y) \, \mathrm{d}y. \tag{15}$$

If an ansatz $\mathcal{K}(x,y) = \delta_x(y)$ is adopted, then (15) reverts to the original one in (10).

- The probing direction $\mathbf{d}(x)$ as "query" is reasonably assumed to have a global dependence on $\phi$

$$\mathbf{d}(x) := \int_\Omega \mathcal{Q}(x,y) \nabla \phi(y) \, \mathrm{d}y. \tag{16}$$

If $\mathcal{Q}(x,y) = \delta_x(y)/\|\nabla\phi(x)\|$, then $\mathbf{d}(x) = \nabla\phi(x)/\|\nabla\phi(x)\|$ which is the choice of the probing direction in (Ikehata, 2000; Ikehata & Siltanen, 2000; Ikehata, 2007).

- In the quantity $R(x)$ in (10), the key is $|\cdot|_Y$ which is assumed to have the following form:

$$|\eta_x|_Y^2 := (\mathcal{V}\eta_x, \eta_x)_{L^2(\partial\Omega)}. \tag{17}$$

In Chow et al. (2014), it is shown that if $\mathcal{V}$ induces a kernel with sharply peaked Gaussian-like distribution, the index function in (10) can achieve maximum values for points inside $D$.

Based on the assumptions from (15) to (17), we derive a matrix representation approximating the new index function on a grid, which accords well with an attention-like architecture. Denote by $\boldsymbol{\phi}_n$: the vector that interpolates $\partial_{x_n}\phi$ at the grid points $\{z_j\}$, $n = 1, 2$.

Here, we sketch the outline of the derivation and present the detailed derivation in Appendix D. We shall discretize the variable $x$ by grid points $z_i$ in (15) and obtain an approximation to the integral:

$$\int_\Omega \mathcal{K}(z_i, y) \partial_{x_n}\phi(y) \, \mathrm{d}y \approx \sum_j \omega_j \mathcal{K}(z_i, z_j) \partial_{x_n}\phi(z_j) =: \mathbf{k}_i^T \boldsymbol{\phi}_n, \tag{18}$$

where $\{\omega_j\}$ are some integration quadrature weights. We then consider (16) and focus on one component $d_n(x)$ of $\mathbf{d}(x)$. With a suitable approximated integral, it can be rewritten as

$$d_n(z_i) \approx \sum_j \omega_j \mathcal{Q}(z_i, z_j) \partial_{x_n}\phi(z_j) =: \mathbf{q}_i^T \boldsymbol{\phi}_n. \tag{19}$$

Note that the self-adjoint positive definite operator $\mathcal{V}$ in (17) can be parameterized by a symmetric positive definite (SPD) matrix denoted by $V$. There exist vectors $\mathbf{v}_{n,i}$ such that $|\eta_{z_i}|_Y^2 \approx \sum_n \boldsymbol{\phi}_n^T \mathbf{v}_{n,i} \mathbf{v}_{n,i}^T \boldsymbol{\phi}_n$. Then, the modified indicator function can be written as

$$\hat{\mathcal{I}}_1^D(z_i) \approx \left\{ \|f - \Lambda_{\sigma_0}g\|_{\partial\Omega}^{-1} \Big( \sum_n \boldsymbol{\phi}_n^T \mathbf{v}_{n,i} \mathbf{v}_{n,i}^T \boldsymbol{\phi}_n \Big)^{-1/2} \right\} \sum_n \boldsymbol{\phi}_n^T \mathbf{q}_i \mathbf{k}_i^T \boldsymbol{\phi}_n. \tag{20}$$

Now, using the notation from Section 3.3, we denote the learnable kernel matrices and an input vector: for $\diamond \in \{Q, K, V\}$, and $\mathbf{u} \in \{\mathbf{q}, \mathbf{k}, \mathbf{v}\}$

$$W^\diamond = \begin{bmatrix} \mathbf{u}_{1,1} & \cdots & \mathbf{u}_{1,M} \\ \mathbf{u}_{2,1} & \cdots & \mathbf{u}_{2,M} \end{bmatrix} \in \mathbb{R}^{2M \times M}, \quad \mathbf{x}_h = \begin{bmatrix} \boldsymbol{\phi}_1 & \boldsymbol{\phi}_2 \end{bmatrix} \in \mathbb{R}^{1 \times 2M}. \tag{21}$$

Then, we can rewrite (20) as

$$[\hat{\mathcal{I}}_1^D(z_i)]_{i=1}^M \approx C_{f,g}(\mathbf{x}_h W^Q * \mathbf{x}_h W^K) / (\mathbf{x}_h W^V * \mathbf{x}_h W^V)^{1/2} \tag{22}$$

where $C_{f,g} = \|f - \Lambda_{\sigma_0}g\|_{\partial\Omega}^{-1}$ is a normalization weight, and both $*$ and $/$ are element-wise. Here, we may define $Q = \mathbf{x}_h W^Q$, $K = \mathbf{x}_h W^K$, and $V = \mathbf{x}_h W^V$ as the query, keys, and values. We can see that the right matrix multiplications (11) in the attention mechanism are low-rank approximations of the ones above. Hence, based on (22), essentially we need to find a function $\mathbf{I}$ resulting in a vector approximation to the true characteristic function $\{\mathcal{I}^D(z_j)\}$

$$\mathbf{I}(Q, K, V) \approx [\mathcal{I}^D(z_i)]_{i=1}^M. \tag{23}$$

Moreover, when there are $L$ data pairs, the data functions $\phi_l$ are generated by computing their harmonic extensions as in (9). Then, each $\phi_l$ is then treated as a channel of the input data $\mathbf{x}_h$.

In summary, the expressions in (22) and (23) reveal that a Transformer may be able to generalize the classical non-parametrized DSM formula further in (10) to non-local learnable kernels. Thus, it may have an intrinsic architectural advantage that handles multiple data pairs. In the subsequent EIT benchmarks, we provide a potential answer to the question (Q2); namely, the attention architecture is better suited for the tasks of reconstruction, as it conforms better with the underlying mathematical structure. The ability to learn global interactions by attention, supported by a non-local kernel interpretation, matches the long-range dependence nature of inverse problems.

## 4 EXPERIMENTS

In this section we present some experimental results to show the quality of the reconstruction. The benchmark contains sampling of inclusions of random ellipses (targets), and the input data has a **single** channel ($L = 1$) of the 2D harmonic extension feature from the 1D boundary measurements. The training uses 1cycle and a mini-batch ADAM for 50 epochs. The evaluated model is taken from the epoch with the best validation metric on a reserved subset. There are several baseline models to compare: the CNN-based U-Nets (Ronneberger et al., 2015; Guo & Jiang, 2020); the state-of-the-art operator learner Fourier Neural Operator (FNO) (Li et al., 2021a) and its variant with a token-mixing layer (Guibas et al., 2022); MultiWavelet Neural Operator (MWO) (Gupta et al., 2021). The Transformer model of interest is a drop-in replacement of the baseline U-Net, and it is named by U-Integral Transformer (UIT). UIT uses the kernel integral inspired attention (11), and we also compare UIT with the linear attention-based Hybrid U-Transformer in Gao et al. (2021), as well as a Hadamard product-based cross-attention U-Transformer in Wang et al. (2022). An ablation study is also conducted by replacing the convolution layers in the U-Net with attention (11) on the coarsest level. For more details of the hyperparameters' setup in the data generation, training, evaluation, network architectures please refer to Section 3.1, Appendix C.1, and Appendix C.2.

The comparison result can be found in Table 1. Because FNO (AFNO, MWO) keeps only the lower modes in spectra, it performs relatively poor in this EIT benchmark where one needs to recover traits that consist of higher modes (sharp boundary edges of inclusion) from lower modes (smooth harmonic extension). Attention-based models are capable to recover "high-frequency target from low-frequency data", and generally outperform the CNN-based U-Nets despite having only $1/3$ of the parameters. Another highlight is that the proposed models are highly robust to noise thanks to the unique PDE-based feature map through harmonic extension. The proposed models can recover the buried domain under a moderately large noise (5%) and an extreme amount of noise (20%) which can be disastrous for many classical methods.

Table 1: Evaluation metrics of the EIT benchmark tests. $\tau$: the normalized relative strength of noises added in the boundary data before the harmonic extension; see Appendix C for details. $L^2$-error and cross entropy: the closer to 0 the better; Dice coefficient: the closer to 1 the better.

| | Relative $L^2$ error | | | Position-wise cross entropy | | | Dice coefficient | | | # params |
|---|---|---|---|---|---|---|---|---|---|---|
| | $\tau = 0$ | $\tau = 0.05$ | $\tau = 0.2$ | $\tau = 0$ | $\tau = 0.05$ | $\tau = 0.2$ | $\tau = 0$ | $\tau = 0.05$ | $\tau = 0.2$ | |
| U-Net baseline | 0.200 | 0.341 | 0.366 | 0.0836 | 0.132 | 0.143 | 0.845 | 0.810 | 0.799 | 7.7m |
| U-Net+Coarse Attn | 0.184 | 0.343 | 0.360 | 0.0801 | 0.136 | 0.147 | 0.852 | 0.807 | 0.804 | 8.4m |
| U-Net big | 0.195 | 0.338 | 0.350 | 0.0791 | 0.133 | 0.138 | 0.850 | 0.812 | 0.805 | 31.0m |
| FNO2d baseline | 0.318 | 0.492 | 0.502 | 0.396 | 0.467 | 0.508 | 0.650 | 0.592 | 0.582 | 10.4m |
| Adaptive FNO2d | 0.323 | 0.497 | 0.499 | 0.391 | 0.466 | 0.471 | 0.635 | 0.595 | 0.592 | 10.9m |
| FNO2d big | 0.386 | 0.482 | 0.501 | 0.310 | 0.465 | 0.499 | 0.638 | 0.601 | 0.580 | 33.6m |
| Multiwavelet NO | 0.275 | 0.390 | 0.407 | 0.152 | 0.178 | 0.192 | 0.715 | 0.694 | 0.688 | 9.8m |
| Hybrid UT | 0.185 | 0.320 | 0.333 | 0.0785 | 0.112 | 0.116 | 0.877 | 0.829 | 0.821 | 11.9m |
| Cross-Attention UT | 0.171 | 0.305 | 0.311 | 0.0619 | 0.105 | 0.109 | 0.887 | 0.840 | 0.829 | 11.4m |
| **UIT+Softmax** (ours) | **0.159** | **0.261** | **0.269** | **0.0551** | **0.0969** | **0.0977** | **0.903** | **0.862** | **0.848** | 11.1m |
| **UIT** (ours) | **0.163** | **0.261** | **0.272** | **0.0564** | **0.0967** | **0.0981** | **0.897** | **0.858** | **0.845** | 11.4m |
| **UIT+**($L$=3) (ours) | **0.147** | **0.250** | **0.254** | **0.0471** | **0.0882** | **0.0900** | **0.914** | **0.891** | **0.880** | 11.4m |

## 5 CONCLUSION

For a boundary value inverse problem, we propose a novel operator learner based on the mathematical structure of the inverse operator and Transformer. The proposed architecture consists of two components: the first one is a harmonic extension of boundary data (a PDE-based feature map), and the second one is a modified attention mechanism derived from the classical DSM by introducing learnable non-local integral kernels. The evaluation accuracy on the benchmark problems surpasses the current widely-used CNN-based U-Net and the best operator learner FNO. This research strengthens the insights that the attention is an adaptable neural architecture that can incorporate a priori mathematical knowledge to design more physics-compatible DNN architectures. However, we acknowledge some limitations: in this study, $\sigma$ to be recovered relies on a piecewise constant assumption. For many EIT applications in medical imaging and industrial monitoring, $\sigma$ may involve non-sharp transitions or even contain highly anisotropic/multiscale behaviors; see Appendix G for more discussion on limitations and possible approaches.

## ACKNOWLEDGMENTS

L. Chen is supported in part by National Science Foundation grants DMS-1913080 and DMS-2012465, and DMS-2132710. S. Cao is supported in part by National Science Foundation grants DMS-1913080 and DMS-2136075. The hardware to perform the experiments are sponsored by NSF grants DMS-2136075, and UMKC School of Science and Engineering computing facilities. No additional revenues are related to this work. The authors would like to thank Ms. Jinrong Wei (University of California Irvine) for the proofreading and various suggestions on the manuscript. The authors would like to thank Dr. Jun Zou (The Chinese University of Hong Kong) and Dr. Bangti Jin (University College London & The Chinese University of Hong Kong) for their comments on inverse problems. The authors also greatly appreciate the valuable suggestions and comments by the anonymous reviewers.

## REPRODUCIBILITY STATEMENT

This paper is reproducible. Experimental details about all empirical results described in this paper are provided in Appendix C. Additionally, we provide the PyTorch (Paszke et al., 2019) code for reproducing our results at `https://github.com/scaomath/eit-transformer`. The dataset used in this paper is available at `https://www.kaggle.com/datasets/scaomath/eletrical-impedance-tomography-dataset`. Formal proofs under a rigorous setting of all our theoretical results are provided in Appendices E-F.

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

# A    TABLE OF NOTATIONS

Table 2: Notations used in an approximate chronological order and their meaning in this work.

| Notation | Meaning |
|:---:|:---|
| $\Omega$ | an underlying spacial domain in $\mathbb{R}^2$ |
| $D$ | a subdomain in $\Omega$ (not necessarily topologically-connected) |
| $\partial D, \partial\Omega$ | $D$'s and $\Omega$'s boundary, 1-dimensional manifolds |
| $\nabla u$ | the gradient vector of a function, $\nabla u(x) = (\partial_{x_1} u(x), \partial_{x_2} u(x))$ |
| $\|\cdot\|_\omega$ | the $L^2$-norm on a region $\omega$ |
| $\|\cdot\| = \|\cdot\|_\Omega$ | the $L^2$-norm on whole domain $\Omega$ |
| $\delta_x$ | the delta function such that $\int_\Omega f(y)\delta_x(y)\,\mathrm{d}y = f(x), \forall f$. |
| $\mathbf{n}$ | the unit outer normal vector on the boundary $\partial\Omega$. |
| $\nabla u \cdot \mathbf{n}$ | normal derivative of $u$, measures the rate of change along the direction of $\mathbf{n}$ |
| $\Lambda_\sigma$ | NtD map from Neumann data $g := \nabla u \cdot \mathbf{n}$ |
| | (how fast the solution changes toward the outward normal direction) to |
| | Dirichlet data $f := u\|_{\partial\Omega}$ (the solution's value along the tangential direction) |
| $H^s(\Omega), s \geq 0$ | the Sobolev space of functions |
| $H^s(\Omega), s < 0$ | the bounded linear functional defined on $H^s(\Omega)$ |
| $H_0^s(\Omega)$ | all $u \in H^s(\Omega)$ such that $u$'s integral on $\Omega$ vanishes |
| $|\cdot|_Y$ | the seminorm defined for functions in $Y$ |

# B    BACKGROUND OF EIT

For EIT, an immediate question is whether $\mathcal{F}^{-1}$ and $\mathcal{F}_L^{-1}$ in (4) and (5) are well-defined, namely whether $\sigma$ can be uniquely determined. In fact, for the case of full measurements ($L = \infty$), the uniqueness for $\mathcal{F}^{-1}$ has been well established, (Brühl, 2001; Hanke & Brühl, 2003; Astala & Päivärinta, 2006; Nachman, 1996; Kohn & Vogelius, 1984; Sylvester & Uhlmann, 1987). It is worthwhile to point out that, in this case, $\sigma$ is not necessarily a piecewise constant function. In (1), we present a simplified case for purposes of illustrating as well as benchmarking. In general, with the full spectral information of the NtD map, $\sigma$ can be uniquely determined as a general positive function.

If infinitely many eigenpairs are known, then the operator itself can be precisely characterized using infinitely many feature channels by Reproducing Kernel Hilbert Space (RKHS) theory, e.g., Mercer (1909); Aronszajn (1950); Minh et al. (2006); Morris (2015); Kadri et al. (2016); Lu et al. (2022). In the context of EIT, this is known as the "full measurement". A more challenging and practical problem is to recover $\sigma$ from only finitely many boundary data pairs. A common practice for the theoretical study of reconstruction using finite measurements is the assumption of $\sigma$ being a piecewise constant function. The task is usually set to recover the shape and location of the inclusion $D$. Otherwise, the problem is too ill-posed. With finite measurements, the uniqueness of the inclusion remains a long-standing theoretical open problem, and it can be only established for several special classes of the inclusion shape, such as the convex cylinders in Isakov & Powell (1990) or convex polyhedrons in Barceló et al. (1994). We refer readers to some counter-examples in Kang & Seo (2001) where a two- or three-dimensional ball may not be identified uniquely by one single measurement if the values of $\sigma_0$ and $\sigma_1$ are unknown.

Furthermore, here we provide one example to illustrate the difficulty in the reconstruction procedure (Pidcock et al., 1995). Let $\Omega$ be a unit circle, let $D$ be a circle with the radius $\rho < 1$, and define

$$\sigma(x) = \begin{cases} 1 & \text{if } \|x\| \geq \rho, \\ \sigma_1 & \text{if } \|x\| < \rho, \end{cases} \tag{24}$$

with $\sigma_1 < 1$ being an arbitrary constant. In this case, the eigenpairs of $\Lambda_\sigma$ can be explicitly calculated

$$\lambda_l = \frac{1}{l}\frac{1 - \rho^{2l}\mu}{1 + \rho^{2l}\mu}, \quad \nu_m = \frac{1}{\sqrt{2\pi}}\cos(l\theta) \text{ or } \frac{1}{\sqrt{2\pi}}\sin(l\theta), \ l = 1, 2, ..., \tag{25}$$

with $\mu = (1 - \sqrt{\sigma_1})/(1 + \sqrt{\sigma_1})$, which are exactly the Fourier modes in a unit circle. In this case, if the set of basis $\{g_l\}_{l=1}^{\infty}$ is just chosen as $\{\cos(\theta), \sin(\theta), \cos(2\theta), \sin(2\theta), ...\}$, the matrix representation $\mathbf{A}_\sigma$ in (3) can be written as an infinite diagonal matrix

$$\mathbf{A}_\sigma \begin{bmatrix} (1-\rho^2\mu)/(1+\rho^2\mu) & 0 & 0 & 0 \\ 0 & (1-\rho^4\mu)/(1+\rho^4\mu) & 0 & 0 \\ 0 & 0 & (1-\rho^4\mu)/(1+\rho^4\mu) & 0 \\ 0 & 0 & 0 & \ddots \end{bmatrix}. \quad (26)$$

Thanks to this special geometry, eigenvalues in (25) can clearly determine $\rho$ and $\sigma_1$ as follows:

$$\rho = \sqrt{\frac{(1-\lambda_2)(1+\lambda_1)}{(1+\lambda_2)(1-\lambda_1)}} \quad \text{and} \quad \mu = \frac{(1-\lambda_1)^2(1+\lambda_2)}{(1+\lambda_1)^2(1-\lambda_2)}. \quad (27)$$

However, in practice, $\mathbf{A}_\sigma$ does not have such a simple structure. Approximating $\mathbf{A}_\sigma$ itself requires a large number of data pairs that are not available in the considered case. Besides, an accurate approximation of the eigenvalues of $\mathbf{A}_\sigma$ is also very expensive. Furthermore, for complex inclusion shapes, two eigenvalues are not sufficient to exactly recover the shape and conductivity values.

## C   EXPERIMENT SET-UP

### C.1   DATA GENERATION AND TRAINING

In the numerical examples, the data generation mainly follows standard practice in theoretical prototyping for solving EIT problems, see e.g.,Chow et al. (2014); Michalikova et al. (2014); Hamilton & Hauptmann (2018); Guo & Jiang (2020); Fan & Ying (2020). For examples, please refer to Figure 2. The computational domain is set to be $\Omega := (-1, 1)^2$, and the two media with the different conductivities are with $\sigma_1 = 10$ (inclusion) and $\sigma_0 = 1$ (background). The inclusions are four random ellipses. The lengths of the semi-major axis and semi-minor axis of these ellipses are sampled from $\mathcal{U}(0.1, 0.2)$ and $U(0.2, 0.4)$, respectively. The rotation angles are sampled from $\mathcal{U}(0, 2\pi)$. There are 10800 samples in the training set, from which 20% are reserved as validation. There are 2000 in the testing set for evaluation.



Figure 2: Randomly selected samples of elliptic inclusion to represent the coefficient $\sigma$ (left 1-4). A Cartesian mesh $\Omega_h$ with a grid point $\{z_j\}$ (right). In computation, discretization of $\mathcal{I}^D$ consists of values taken as 1 at the mesh points of $\Omega_h$ inside $D$ and 0 at others.

The noise $\xi = \xi(x)$ below (10) is assumed to be

$$\xi(x) = (f(x) - \Lambda_{\sigma_0} g(x))\tau G(x) \quad (28)$$

where $\tau$ specifies the relative strength of noise, and $G(x)$ is a normal Gaussian distribution independent with respect to $x$. As $\xi(x)$ is merely pointwise imposed, the boundary data can be highly rough, even if the ground truth $f(\cdot) - \Lambda_{\sigma_0} g(\cdot)$ is chosen to be smooth. Nevertheless, the harmonic extension makes the noise from boundary data have a minimal impact on the overall reconstruction, thanks to the smoothing property of the inverse of the Laplacian operator $(-\Delta)^{-1}$; for example, please refer to Figure 3. In data generation, the harmonic extension feature map is approximated by finite element methods incorporating stencil modification near the inclusion interfaces (Guo & Lin, 2019; Guo et al., 2019). Similar data augmentation practices using internal data can be found in Nachman et al. (2007); Bal (2013); Jin et al. (2022a).

Thanks to the position-wise binary nature of $\mathcal{I}^D$, another choice of the loss function during training can be the binary cross entropy $\mathcal{L}(\cdot, \cdot)$, applied for a function in $\mathbb{P}$, to measure the distance between the ground truth and the network's prediction

$$\mathcal{L}(\mathbf{p}_h, \mathbf{u}_h) := -\sum_{z \in \mathcal{M}} \big(\mathbf{p}_h(z) \ln(\mathbf{u}_h(z)) + (1 - \mathbf{p}_h(z)) \ln(1 - \mathbf{u}_h(z))\big). \quad (29)$$

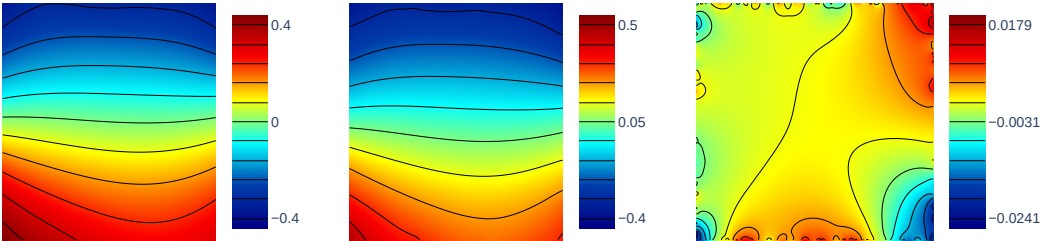

Figure 3: The harmonic extensions $\phi_l$ with zero noise on boundary (left); the harmonic extensions $\tilde{\phi}_l$ with 20% Gaussian noise on boundary (middle). Their pointwise difference (right). $\|\phi_l - \tilde{\phi}_l\| / \|\phi_l\| = 0.0203$.

Thanks to the Pinsker inequality (e.g., see (Cover, 1999, Section 11.6)), $\mathcal{L}(\mathbf{p}_h, \mathbf{u}_h)$ serves as a good upper bound for the square of the total variation, which can be further bounded below by the $L^2$-error given the boundedness of the position-wise value.

The training uses `1cycle` (Smith & Topin, 2019) learning rate strategy with a warm-up phase. A mini-batch ADAM iterations are run for 50 epochs with no extra regularization, such as weight decay. The evaluated model is taken from the epoch that has the best validation metric. The learning rate starts and ends with $10^{-3} \cdot lr_{\max}$, and reaches the maximum of $lr_{\max}$ at the end of the 10-th epoch. The $lr_{\max} = 10^{-3}$. The result demonstrated is obtained from fixing the random number generator seed. Figure 6 shows the testing results for a randomly chosen sample. All models are trained on an RTX 3090 or an A4000. The codes to replicate the experiments are open-source and publicly available. [1].

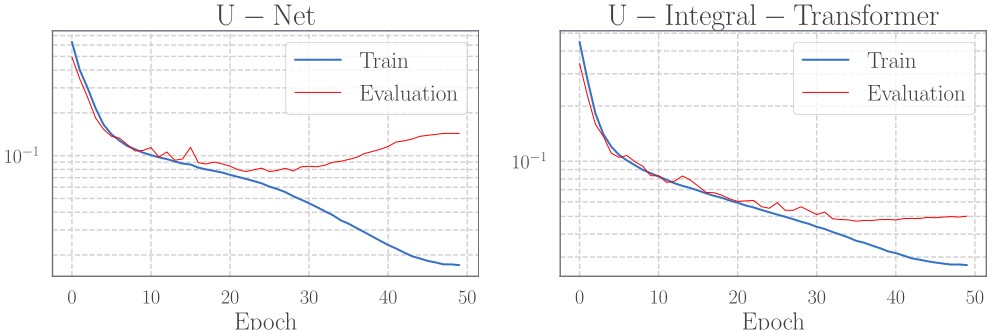

Figure 4: The left: the training-testing pixel-wise binary cross entropy convergence for the CNN-based U-Net with 31 million parameters, a clear overfitting pattern is shown. The right: the training-testing convergence for the attention-based U-Transformer with 11.4 million parameters.

## C.2 NETWORK ARCHITECTURE

The difference in architectural hyperparameters, together with training and evaluation costs comparison for all the models compared in the task of EIT reconstruction can be found in Table 3.

**U-Integral Transformer**

- *Overall architecture.* The U-Integral-Transformer architecture is a drop-in replacement of the standard CNN-based U-Net baseline model (7.7m) in Table 1. The CNN-based U-Net is used in DL-based approaches for boundary value inverse problem in Guo & Jiang (2020); Guo et al. (2021); Le et al. (2022). One of the novelties is that the input is a tensor that concatenates different measurement matrices as different channels, and a similar practice can be found in Brandstetter et al. (2023). Same with the baseline U-Net, the UIT has three downsampling layers as the encoder (feature extractor). The downsampling layers map $m \times m$ latent representations to $m/2 \times m/2$, and

---

[1] https://github.com/scaomath/eit-transformer

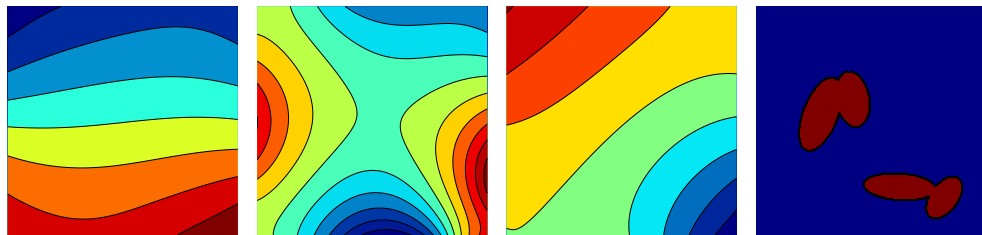

Figure 5: The harmonic extension feature map $\phi_l$ (left 1-3 as different channels' inputs to the neural network) corresponding to a randomly chosen sample's inclusion map (right). No visible relevance shown with the ground truth. The layered heatmap appearance is adopted in plotly contour only for aesthetics purposes, no edge-like nor layer-like features can be observed from the actual harmonic extension feature map input.

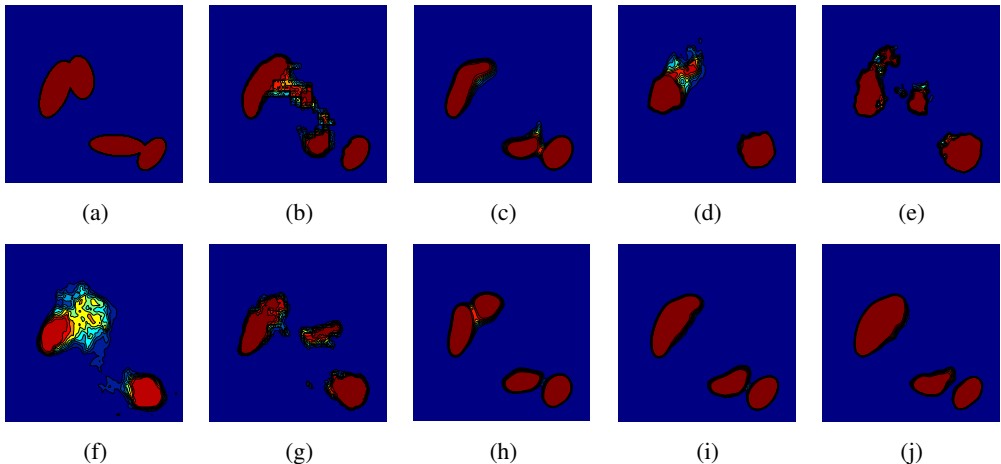

Figure 6: The neural network evaluation result for the inclusion in Figure 5 using various models. A model's input is the left most function in Figure 5 if this model uses a single channel, or all left three in Figure 5 if a model uses 3-channel input. (a) Ground truth inclusion in Figure 5; (b) U-Net baseline (7.7m) prediction; (c) U-Net big (31m) prediction with 3 channels; (d) Fourier Neural Operator (10.4m) prediction with 1 channel; (e) Fourier Neural Operator big (33m) prediction with 1 channel; (f) Adaptive Fourier Neural Operator (10.7m) prediction with 1 channel; (g) Multiwavelet Neural Operator (9.8m) prediction with 1 channel; (h) Hybrid UT with a linear attention (10.13m) prediction with 1 channel; (i) UIT (11.4m) prediction with 1 channel; (j) UIT (11.4m) prediction with 3 channels.

expand the number of channels from $C$ to $2C$. To leverage the "basis ⇔ channel" interpretation and the basis update nature of the attention mechanism in Section 3.3, the proposed attention block is first added on the coarsest grid, which has the most number of channels. UIT has three upsampling layers as the decoder (feature selector), which map $m/2 \times m/2$ latent representations to $m \times m$, and shrink the number of channels from $2C$ to $C$. In these upsampling layers, attention blocks are applied on each cross-layer propagation to compute the interaction between the latent representations on both coarse and fine grids (see below). Please refer to Figure 8 for a high-level encoder-decoder schematic.

- *Double convolution block.* The double convolution block is modified from that commonly seen in Computer Vision (CV) models, such as ResNet (He et al., 2016). We modify this block such that upon being used in an attention block, the batch normalization (Ioffe & Szegedy, 2015) can be replaced by the layer normalization (Ba et al., 2016), which can be understood as a learnable approximation to the Gram matrices' inverse by a diagonal matrix.

- *Positional embedding.* At each resolution, the 2D Euclidean coordinates of an $m \times m$ regular Cartesian grid are the input of a channel expansion through a fixed learnable linear layer and are

then added to each latent representation. This choice of positional embedding enables a bilinear interpolation between the coarse and fine grids or vice versa (see below).

- *Mesh-normalized attention.* The scaled dot-product attention in the network is chosen to be the integral kernel attention in (11) with a mesh-based normalization. Please refer to Figure 7 for a diagram in a single attention head.

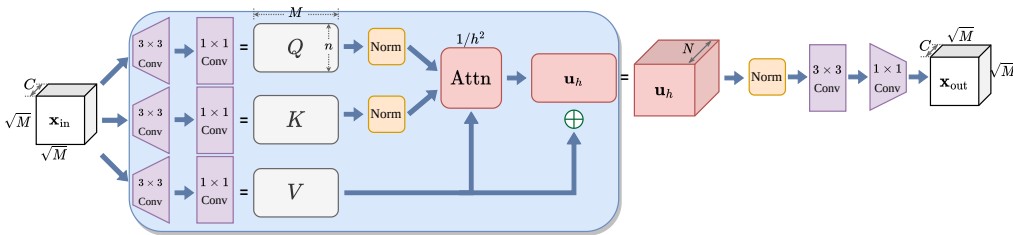

Figure 7: Detailed flow of the modified 2D attention-based encoder layer using (11). $C$: the number of channels in the input, $N$: the number of expanded channels (for the basis expansion interpretation in Theorem 2).

- *Interpolation.* Instead of max pooling used in the standard CNN-based U-Net, we opt for a bilinear interpolation on the Cartesian grid to map a latent representation from the fine grid to the coarse grid or vice versa. Note that, in the upsampling layers, the interpolation's outputs are directly inputted into an attention block that computes the interaction of latent representations between coarse and fine grids (see below).

- *Coarse-fine attention in up blocks.* A modified attention in Section 3.3 with a pre-inner-product normalization replaces the convolution layer on the coarsest level. The skip connection from the encoder latent representations to the ones in the decoder are generated using an architecture similar to the cross attention used in Petit et al. (2021). $Q$ and $K$ are generated from the latent representation functions on the same coarser grids. As such, the attention kernel to measure the interaction between different channels is built from the coarse grid. $V$ is associated with a finer grid. Compared with the one in Petit et al. (2021), the modified attention in our method is inspired by the kernel integral for a PDE problem. Thus, it has (1) no softmax normalization or (2) no Hadamard product-type skip connection.

**Other operator learners compared**

- *Fourier Neural Operator (FNO) and variants.* Fourier Neural Operator (FNO2d) learns convolutional filters in the frequency domain for some pre-selected modes, efficiently capturing globally-supported spatial interactions for these modes. The weight filter in the frequency domain multiplies with the lowest modes in the latent representations (four corners in the FFT). The Adaptive Fourier Neural Operator (AFNO2d) adds a token-mixing layer, as seen in Figure 2 in Guibas et al. (2022), appending every spectral convolution layer in the baseline FNO2d model.

- *Multi-Wavelet Neural Operator (MWO).* The MultiWavelet Neural Operator (MWO) is proposed in Gupta et al. (2021), which introduces a multilevel structure into the FNO architecture. MWO still follows FNO's practice on each level by pre-selecting the lowest modes.

## D FROM DSM TO TRANSFORMER

This section gives a more detailed presentation of how the attention-like operator is derived from the DSM ansatz (15) with learnable kernels.

We begin with recalling the original indicator function from (10):

$$\mathcal{I}^D(x) := C_{f,g} \frac{\mathbf{d}(x) \cdot \nabla \phi(x)}{|\eta_x|_{H^s(\partial\Omega)}}, \tag{30}$$

where $C_{f,g} = \|f - \Lambda_{\sigma_0} g\|_{L^2(\partial\Omega)}^{-1}$ is a constant, and the equations of the functions $\phi$ and $\eta_x$ are :

$$-\Delta\phi = 0 \quad \text{in} \quad \Omega, \quad \mathbf{n} \cdot \nabla\phi = (f - \Lambda_{\sigma_0} g) + \xi \quad \text{on} \quad \partial\Omega, \quad \int_{\partial\Omega} \phi \, ds = 0, \tag{31}$$

Table 3: The detailed comparison of the networks used in this study. For U-Net-based neural networks, the channel/width is the number of the base channels on the finest grid after the initial channel expansion. A `torch.cfloat` type parameter entry counts as two parameters. GFLOPs: Giga FLOPs for 1 backpropagation (BP) performed for a batch of 8 samples recorded the PyTorch `autograd` profiler for 1 BP averaging from 100 BPs. Eval: number of instances per second.

| | Architectures | | | | Training/Evaluation cost | | # params |
|---|---|---|---|---|---|---|---|
| | layers | channel/width | modes | norm | GFLOPs | eval | |
| U-Net | 7 | 64 | N/A | batch | 140.6 | 298.6 | 7.70m |
| U-Net big | 9 | 64 | N/A | batch | 184.1 | 273.4 | 31.04m |
| FNO2d | 6 | 48 | 14 | FFT | 196.5 | 235.4 | 10.86m |
| AFNO2d | 6 | 48 | 14 | FFT | 198.4 | 151.0 | 10.88m |
| MWO2d | 4 | 64 | 12 | Legendre | 1059 | 59.6 | 9.81m |
| Hybrid UT | 7 | 32 | N/A | batch | 427.5 | 114.0 | 10.13m |
| Cross-Attn UT | 7 | 64 | N/A | layer | 658.9 | 103.3 | 11.42m |
| UIT (ours) | 7 | 64 | N/A | layer | 658.3 | 104.8 | 11.43m |

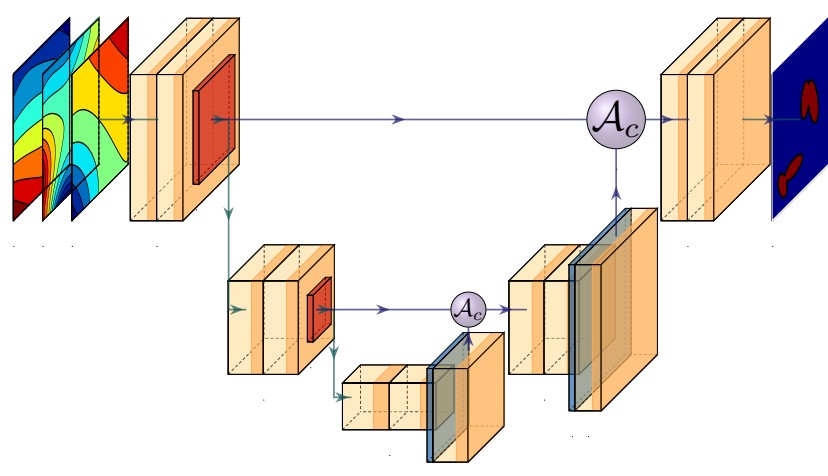

Figure 8: A simplified schematic of the U-Integral-Transformer that follows the standard U-Net. The input is a tensor concatenating the discretizations of $\phi$ and $\nabla\phi$. The output is the approximation to the index function $\mathcal{I}^D$. ▨: $3 \times 3$ convolution + ReLU; ▨: layer normalization or batch normalization; ▨: bilinear interpolations from the fine grid to the coarse grid; ▨: cross attention $\mathcal{A}_c$ that uses the latent representations on a coarse grid to compute interactions to produce the latent representations on a finer grid; ▨: input and output discretized functions in certain Hilbert spaces. The TikZ source code to produce this figure is modified from the examples in Iqbal (2018).

$$-\Delta\eta_x = -\mathbf{d}(x) \cdot \nabla\delta_x \quad \text{in} \quad \Omega, \quad \mathbf{n} \cdot \nabla\eta_x = 0 \quad \text{on} \quad \partial\Omega, \quad \int_{\partial\Omega} \eta_x \, \mathrm{d}s = 0. \tag{32}$$

Chow et al. (2014) shows that $\mathcal{I}^D(x)$ can be written as a sum of Gaussian-like distributions that attain maximum values for $x \in D$ or close to $D$. However, the accuracy is much limited by empirical choices of quantities in (30), for example, $\mathbf{d}(x) = \nabla\phi(x)/\|\nabla\phi(x)\|$ and $s = 3/2$. In addition, the frequency of the boundary data plays an important role in reconstruction; see the derivation in Chow et al. (2014, Section 5) for circular $\Omega$. What is more, such a simple formula can be derived only for a single data pair.

Henceforth, these "empirical chosen quantities" are made to be learnable from data by introducing two undetermined kernels $\mathcal{K}(x, y), \mathcal{Q}(x, y)$, and a self-adjoint positive definite linear operator $\mathcal{V}$, the modified indicator function is written as

$$\hat{\mathcal{I}}_1^D(x) := C_{f,g} \frac{\int_\Omega \mathbf{d}(x) \cdot \mathcal{K}(x, y)\nabla\phi(y) \, \mathrm{d}y}{|\eta_x|_Y}. \tag{33}$$

with

$$\mathbf{d}(x) := \int_\Omega \mathcal{Q}(x, y)\nabla\phi(y) \, \mathrm{d}y. \tag{34}$$

$$|\eta_x|_Y^2 := (\mathcal{V}\eta_x, \eta_x)_{L^2(\partial\Omega)}. \tag{35}$$

Applying certain quadrature rule to (33) with the quadrature points $z_i$, i.e., the grid points of $\Omega_h$, and weights $\{\omega_j\}$, we obtain an approximation to the integral:

$$\int_\Omega \mathcal{K}(z_i, y)\partial_{x_n}\phi(y)\,\mathrm{d}y \approx \sum_j \omega_j \mathcal{K}(z_i, z_j)\partial_{x_n}\phi(z_j) =: \mathbf{k}_i^T\boldsymbol{\phi}_n, \tag{36}$$

i.e., $\mathbf{q}_i^T$ is the vector of $[\omega_j\mathcal{K}(z_i, z_j)]_j$. For (34), we consider one component $d_n(x)$ of $\mathbf{d}(x)$. With the same rule to compute the integral, (16) can be written as

$$d_n(z_i) \approx \sum_j \omega_j \mathcal{Q}(z_i, z_j)\partial_{x_n}\phi(z_j) =: \mathbf{q}_i^T\boldsymbol{\phi}_n, \tag{37}$$

i.e., $\mathbf{q}_i^T$ is the vector of $[\omega_j\mathcal{Q}(z_i, z_j)]_j$. Next, we proceed to express $|\eta_{z_i}|_Y$ by discretizing the variational form in (32) using a linear finite element method (FEM). Applying integration by parts, the weak form of (32) at $x = z_i$ is

$$\int_\Omega \nabla\eta_{z_i} \cdot \nabla\psi\,\mathrm{d}y = \int_\Omega -\mathbf{d}(z_i) \cdot \nabla\delta_{z_i}(y)\psi(y)\,\mathrm{d}y = -\mathbf{d}(z_i) \cdot \nabla\psi(z_i), \tag{38}$$

for any test function $\psi \in H_0^1(\Omega)$. Here, we let $\{\psi^{(j)}\}_{j=1}^M$ be the collection of the finite element basis functions, and for the fixed $z_i$ we further let $\boldsymbol{\psi}_{n,i}$ be the vector approximating $[\partial_{x_n}\psi^{(j)}(z_i)]_{j=1}^M$. Denote $\boldsymbol{\eta}_i$ as the vector approximating $\{\eta_{z_i}(z_j)\}_{j=1}^M$. Introduce the matrices

- $B$: the finite element/finite difference discretization of $-\Delta$ on $\Omega_h$ (a discrete Laplacian) coupled with the Neumann boundary condition and the zero integral normalization condition in (9).
- $R$: the matrix that projects a vector defined at interior grids to the one defined on $\partial\Omega$.

Then, the finite element discretization of (38) yields the linear system:

$$B\boldsymbol{\eta}_i = \sum_n d_n(z_i)\boldsymbol{\psi}_{n,i} \approx \sum_n \boldsymbol{\psi}_{n,i}\mathbf{q}_i^T\boldsymbol{\phi}_n, \tag{39}$$

where we have used (37). Then, the trace of $\eta_{z_i}$ on $\partial\Omega$ admits the following approximation

$$\bar{\boldsymbol{\eta}}_i := \eta_{z_i}|_{\partial\Omega} \approx \sum_n RB^{-1}\boldsymbol{\psi}_{n,i}\mathbf{q}_i^T\boldsymbol{\phi}_n. \tag{40}$$

Now, we can discretize (35). Note that the trace of a linear finite element space on $\partial\Omega$ is still a continuous piecewise linear space, defined as $S_h(\partial\Omega)$. Then, the self-adjoint positive definite operator $\mathcal{V}$ can be parameterized by a symmetric positive definite (SPD) matrix denoted by $V$ operating on the space $S_h(\partial\Omega)$. We can approximate $|\eta_{z_i}|_Y^2$ as

$$|\eta_{z_i}|_Y^2 \approx \bar{\boldsymbol{\eta}}_i^T V \bar{\boldsymbol{\eta}}_i \approx \sum_n \boldsymbol{\psi}_{n,i}^T B^{-1}R^T V RB^{-1}\boldsymbol{\psi}_{n,i}\boldsymbol{\phi}_n^T\mathbf{q}_i\mathbf{q}_i^T\boldsymbol{\phi}_n \tag{41}$$

where $\boldsymbol{\psi}_{n,i}^T B^{-1}R^T V RB^{-1}\boldsymbol{\psi}_{n,i} \geq 0$ as $V$ is SPD. Define

$$\mathbf{v}_{n,i} = (\boldsymbol{\psi}_{n,i}^T B^{-1}R^T V RB^{-1}\boldsymbol{\psi}_{n,i})^{1/2}\mathbf{q}_i. \tag{42}$$

This can be considered another learnable vector since the coefficient of $\mathbf{q}_i$ comes from the learnable matrix $V$. Then, (41) reduces to

$$|\eta_{z_i}|_Y^2 \approx \sum_n \boldsymbol{\phi}_n^T\mathbf{v}_{n,i}\mathbf{v}_{n,i}^T\boldsymbol{\phi}_n \tag{43}$$

Putting (36), (37) and (43) into (15), we have

$$\hat{\mathcal{I}}_1^D(z_i) \approx \left\{ \|f - \Lambda_{\sigma_0}g\|_{L^2(\partial\Omega)}^{-1}\Big(\sum_n \boldsymbol{\phi}_n^T\mathbf{v}_{n,i}\mathbf{v}_{n,i}^T\boldsymbol{\phi}_n\Big)^{-1/2} \right\} \sum_n \boldsymbol{\phi}_n^T\mathbf{q}_i\mathbf{k}_i^T\boldsymbol{\phi}_n. \tag{44}$$

Now, using the notation in (21), we get the desired representation (22).

# E  PROOF OF THEOREM 1

**Lemma 1.** *Suppose the boundary data $g_l$ is the eigenfunction of $\Lambda_\sigma - \Lambda_{\sigma_0}$ corresponding to the $l$-th eigenvalue $\lambda_l$, and let $\phi_l$ be the data functions generated by harmonic extensions*

$$-\Delta\phi_l = 0 \quad in \quad \Omega, \quad \mathbf{n}\cdot\nabla\phi_l = (f_l - \Lambda_{\sigma_0} g_l) = (\Lambda_\sigma - \Lambda_{\sigma_0})g_l \quad on \quad \partial\Omega, \quad \int_{\partial\Omega}\phi_l \, \mathrm{d}s = 0, \quad (45)$$

*where $l = 1, 2, \cdots$. Let $\mathbf{d}$ be an arbitrary unit vector in $\mathbb{R}^2$, define a function*

$$\Theta_L(x) = \sum_{l=1}^{L}\frac{(\mathbf{d}\cdot\nabla\phi_l(x))^2}{\lambda_l^3}. \quad (46)$$

*Then, there holds*

$$\lim_{L\to\infty}\Theta_L(x) = \begin{cases} \infty, & if \ x \notin D, \\ a \ finite \ constant, & if \ x \in D. \end{cases} \quad (47)$$

*Proof.* See Theorem 4.1 in Guo & Jiang (2020), and also see Brühl (2001); Hanke & Brühl (2003). □

**Theorem 1** (A finite-dimensional approximation of the index function). *Suppose the boundary data $g_l$ is the eigenfunction of $\Lambda_\sigma - \Lambda_{\sigma_0}$ corresponding to the $l$-th eigenvalue $\lambda_l$, and let $\phi_l$ be the data functions generated by harmonic extensions given in (9). Define the space:*

$$\widetilde{\mathbb{S}}_L = \mathrm{Span}\{\partial_{x_1}\phi_l \ \partial_{x_2}\phi_l : \ l = 1, ..., L\}, \quad (48)$$

*and the dictionary:*

$$\mathbb{S}_L = \{a_1 + a_2\arctan(a_3 v) : \ v \in \widetilde{\mathbb{S}}_L, \ a_1, a_2, a_3 \in \mathbb{R}\}. \quad (49)$$

*Then, for any $\epsilon > 0$, we con construct an index function $\mathcal{I}_L^D \in \mathbb{S}_L$ s.t.*

$$\sup_{x\in\Omega}|\mathcal{I}^D(x) - \mathcal{I}_L^D(x)| \le \epsilon \quad (50)$$

*provided $L$ is large enough.*

*Proof.* Consider the function $\Theta_L(x)$ from Lemma 1. As $\Theta_L(x) > 0$, it is increasing with respect to $L$. Then, there is a constant $\rho$ such that $\rho > \Theta_L(x), \forall x \in D$. Given any $\epsilon > 0$, there is an integer $L$ such that $\Theta_L(x) > 4\rho\epsilon^{-2}/\pi^2, \forall x \notin D$. Define

$$\mathcal{I}_L^D(x) = 1 - \frac{2}{\pi}\arctan\left(\frac{\pi\epsilon}{2\rho}\Theta_L(x)\right) \quad (51)$$

Note the fundamental inequality $z > \arctan(z) \ge \frac{\pi}{2} - z^{-1}, \forall z > 0$. Then, if $x \in D$, there holds

$$|\mathcal{I}^D(x) - \mathcal{I}_L^D(x)| = \frac{2}{\pi}\arctan\left(\frac{\pi\epsilon}{2\rho}\Theta_L(x)\right) < \frac{\epsilon}{\rho}\Theta_L(x) < \epsilon$$

if $x \notin D$, there holds

$$|\mathcal{I}^D(x) - \mathcal{I}_L^D(x)| = 1 - \frac{2}{\pi}\arctan\left(\frac{\pi\epsilon}{2\rho}\Theta_L(x)\right) \le \frac{4\rho}{\pi^2\epsilon\Theta_L(x)} < \epsilon.$$

Therefore, the function in (51) fulfills (50).

□

# F  PROOF OF THEOREM 2

In presenting Theorem 2 in Section 3.3, we use the term "multiplicative" to describe the fact that two latent representations are multiplied in the attention mechanism. In contrast, no such operation exists in, e.g., a pointwise FFN or a convolution layer. Heuristically speaking, the main result in Theorem 2 states that the output latent representations can be of a higher "frequency" than the input if the neural network architecture has "multiplicative" layers in it. The input latent representations are discretizations of certain functions, and they are combined using matrix dot product as the one used in attention. Suppose that this discretization can represent functions of such a frequency with a certain approximation error, the resulting matrix/tensor can be an approximation of a function with a higher frequency than the existing latent representation under the same discretization. Please see Figure 9 and Figure 10 for empirical evidence of this phenomenon for the latent representations: with completely smooth input (harmonic extensions), e.g., see Figure 5, attention-based learner can generate latent representations with multiple peaks and valleys.

**Theorem 2** (Frequency-bootstrapping for multiplicative neural architectures). *Consider $\Omega = (0, \pi)$ which has a uniform discretization of $\{z_i\}_{i=1}^M$ of size $h$, and $v(x) = \sin(ax)$ for some $a \in \mathbb{Z}^+$. Let $N := a - 1 \geq 1$ be the number of channels in the attention layer of interest, assume that (i) the current latent representation $\mathbf{p}_h \in \mathbb{R}^{M \times N}$ consists of the discretization of the first $N$ Legendre polynomials $\{p_j(\cdot)\}_{j=1}^N$ such that $(\mathbf{p}_h)_{ij} = p_j(z_i)$, (ii) $\mathbf{p}_h$ is normalized and the normalization weights $\alpha \equiv 1$ in (14), (iii) the discretization satisfies that $|\sum_{i=1}^M h f(z_i) - \int_\Omega f(x)\,\mathrm{d}x| \leq C\,h$. Then, there exists a set of attention weights $\{W^Q, W^K, W^V\}$ such that for $u(x) = \sin(a'x)$ with $\mathbb{Z}^+ \ni a' > a$*

$$\|\tilde{u} - u\|_{L^2(\Omega)} \leq C \max\{h, \|\varepsilon\|_{L^\infty(\Omega)}\}, \tag{52}$$

*where $\tilde{u}$ and $\tilde{\kappa}(\cdot, \cdot)$ are defined as the output of and the kernel of the attention formulation in (14), respectively; $\varepsilon(x) := \|\kappa(x, \cdot) - \tilde{\kappa}(x, \cdot)\|_{L^2(\Omega)}$ is the error function for the kernel approximation.*

*Proof.* Without loss of generality, it is assumed that $a' = a + 1$. The essential technical tools used suggest the validity for any $a' > a > 0$ ($a', a \in \mathbb{Z}^+$). Consider a simple non-separable smooth kernel function

$$\kappa(x, z) := \sin((a + 1)(x - z)), \tag{53}$$

it is straightforward to verify that for $v(x) := \sin(ax)$, we have $c_1 = 2a/(2a + 1)$

$$\int_\Omega \kappa(x, z) v(x)\,\mathrm{d}x = \int_\Omega \sin\big((a + 1)(x - z)\big) \sin(ax)\,\mathrm{d}x = c_1 \sin\big((a + 1)z\big) =: c_1 u(z). \tag{54}$$

As is shown, it suffices to show that the matrix multiplication (a separable kernel) in the attention mechanism approximates this non-separable kernel with an error related to the number of channels. To this end, taking the Taylor expansion, centered at a $z_0 \in \Omega$ of $\kappa(x, \cdot)$ with respect to the second variable at each $x \in \Omega$, we have

$$\kappa_N(x, z) := \sum_{l=1}^N \frac{(z - z_0)^{l-1}}{(l-1)!} \frac{\partial^{l-1} \kappa}{\partial z^{l-1}}(x, z_0). \tag{55}$$

It is straightforward to check that

$$\|\kappa(x, \cdot) - \kappa_N(x, \cdot)\|_{L^2(\Omega)} \leq c_1 \frac{\sqrt{(\pi - z_0)^{2N+1} + z_0^{2N+1}}}{N!\sqrt{2N+1}} \left\| \frac{\partial^N \kappa}{\partial z^N}(x, \cdot) \right\|_{L^\infty(\Omega)}. \tag{56}$$

By the assumptions on $\kappa(\cdot, \cdot)$, and a straightforward computation we have

$$\|\kappa(x, \cdot) - \kappa_N(x, \cdot)\|_{L^2(\Omega)} \leq c_2 \frac{(\pi)^N}{N!\sqrt{N}} \left\| \frac{\partial^N \kappa}{\partial z^N}(x, \cdot) \right\|_{L^\infty(\Omega)}. \tag{57}$$

Next, let $q_l(z) := (z - z_0)^{l-1}/(l-1)!$ and $k_l(x) := \partial^{l-1}\kappa/\partial z^{l-1}(x, z_0)$ for $1 \leq l \leq N$, i.e., they form a Pincherle-Goursat (degenerate) kernels (Kress, 1999, Chapter 11)

$$\kappa_N(x, z) = \sum_{l=1}^N q_l(x) k_l(z). \tag{58}$$

By this choice of the latent representation space being the first $N$ Legendre polynomials, $q_l \in \mathbb{Y} := \text{span}\{p_j\}$, thus there exists a set of weights $\{w_l^Q \in \mathbb{R}^l\}_{l=1}^N$ corresponding to each channel, such that

$$q_l(\cdot) = \sum_{j=1}^N w_{l,j}^Q p_j(\cdot) =: \tilde{q}_l(\cdot). \tag{59}$$

This is to say, $Q = \mathbf{p}_h W^Q \in \mathbb{R}^{M \times N}$ with the $l$-th column of $Q$ being the discretization of $\tilde{q}_l(\cdot)$.

For the key matrix, by standard polynomial approximation since $\mathbb{Y} \simeq \mathbb{P}_N(\Omega)$, there exists a set of weights $\{w_l^K \in \mathbb{R}^N\}_{l=1}^N$, such that

$$k_l(\cdot) \approx \sum_{j=1}^N w_{l,j}^K p_j(\cdot) =: \tilde{k}_l(\cdot), \tag{60}$$

i.e., $K = \mathbf{p}_h W^K \in \mathbb{R}^{M \times N}$ with the $l$-th column of $K$ being the discretization of $\tilde{k}_l(\cdot)$. Moreover, it can approximate $k_l(\cdot)$ with the following estimate

$$\|k_l(\cdot) - \tilde{k}_l(\cdot)\|_{L^2(\Omega)} \leq c_3 \frac{\pi^N}{2^N (N+1)^N} |k_l(\cdot)|_{H^N(\Omega)}. \tag{61}$$

Similarly, without loss of generality, we choose $v(\cdot) := v_1(\cdot)$, which is concatenated to $V$ such that it occupies the first channel of $V$ defined earlier, we have $\{w_l^V \in \mathbb{R}^N\}$ such that

$$v(\cdot) \approx \sum_{j=1}^N w_{1,j}^K p_j(\cdot) =: \tilde{v}(\cdot) \tag{62}$$

and

$$\|v(\cdot) - \tilde{v}(\cdot)\|_{L^2(\Omega)} \leq c_4 \frac{\pi^N}{2^N (N+1)^N} |v(\cdot)|_{H^N(\Omega)}. \tag{63}$$

Now, to approximate the frequency bootstrapping in (54), define

$$\tilde{u}(z) := \int_\Omega \tilde{\kappa}_N(x, z)\tilde{v}(x)\,\mathrm{d}x, \quad \text{with } \tilde{\kappa}_N(x, z) := \sum_{l=1}^N \tilde{q}_l(x)\tilde{k}_l(z). \tag{64}$$

Then, we have for any $z \in \Omega$

$$\begin{aligned}
u(z) - \tilde{u}(z) &= \int_\Omega \kappa(x, z)v(x)\,\mathrm{d}x - \int_\Omega \kappa_N(x, z)\tilde{v}(x)\,\mathrm{d}x \\
&= \int_\Omega \big(\kappa(x, z) - \tilde{\kappa}_N(x, z)\big)v(x)\,\mathrm{d}x + \int_\Omega \kappa_N(x, z)\big(v(x) - \tilde{v}(x)\big)\,\mathrm{d}x.
\end{aligned} \tag{65}$$

Thus, we have

$$\begin{aligned}
\|u - \tilde{u}\|_{L^\infty(\Omega)} &\leq \max_{z \in \Omega} \left\{ \int_\Omega \big|\big(\kappa(x, z) - \tilde{\kappa}_N(x, z)\big)v(x)\big|\,\mathrm{d}x + \int_\Omega \big|\kappa_N(x, z)\big(v(x) - \tilde{v}(x)\big)\big|\,\mathrm{d}x \right\} \\
&\leq \max_{z \in \Omega} \Big\{ \underbrace{\|\kappa(x, \cdot) - \tilde{\kappa}_N(x, \cdot)\|_{L^2(\Omega)}}_{(*)} \|v\|_{L^2(\Omega)} + \|\tilde{\kappa}_N(x, \cdot)\|_{L^2(\Omega)} \|v - \tilde{v}\|_{L^2(\Omega)} \Big\}.
\end{aligned}$$

$$(66)$$

Now, by triangle inequality, $q_l = \tilde{q}_l$, the definitions above (53) implying $|k_l(\cdot)|_{H^N(\Omega)} \leq c\,2^N$ and $\|q_l\|_{L^2(\Omega)} \leq c\pi^N/(\sqrt{N}N!)$, and the estimate in (61)

$$\begin{aligned}
(*) &\leq \|\kappa(x, \cdot) - \kappa_N(x, \cdot)\|_{L^2(\Omega)} + \|\kappa_N(x, \cdot) - \tilde{\kappa}_N(x, \cdot)\|_{L^2(\Omega)} \\
&\leq \|\kappa(x, \cdot) - \kappa_N(x, \cdot)\|_{L^2(\Omega)} + \sum_{l=1}^N \|q_l\|_{L^2(\Omega)} \|k_l - \tilde{k}_l\|_{L^2(\Omega)} \\
&\leq c_5 \left( \frac{(2\pi)^N}{N!\sqrt{N}} + c_6 \frac{\pi^N}{2^N(N+1)^N} \sum_{l=1}^N \|q_l\|_{L^2(\Omega)} |k_l(\cdot)|_{H^N(\Omega)} \right) \\
&\leq c_6 \frac{(2\pi)^N}{N!\sqrt{N}}.
\end{aligned} \tag{67}$$

Notice this is the same order with the estimate of $\|\kappa(x,\cdot) - \kappa_N(x,\cdot)\|_{L^2(\Omega)}$. For the term $\|\tilde{\kappa}_N(x,\cdot)\|_{L^2(\Omega)}$, a simple triangle inequality trick can be used:

$$
\begin{aligned}
\|\tilde{\kappa}_N(x,\cdot)\|_{L^2(\Omega)} &\leq \|\kappa_N(x,\cdot)\|_{L^2(\Omega)} + \|\kappa_N(x,\cdot) - \tilde{\kappa}_N(x,\cdot)\|_{L^2(\Omega)} \\
&\leq \|\kappa(x,\cdot)\|_{L^2(\Omega)} + \|\kappa_N(x,\cdot) - \tilde{\kappa}_N(x,\cdot)\|_{L^2(\Omega)},
\end{aligned}
\tag{68}
$$

which can be further estimated by reusing the argument in (67).

Lastly, using the following argument and the estimate for $\|u - \tilde{u}\|_{L^\infty(\Omega)}$ yield the desired result:

$$
\|u - \tilde{u}\|^2_{L^2(\Omega)} \leq \|u - \tilde{u}\|_{L^1(\Omega)}\|u - \tilde{u}\|_{L^\infty(\Omega)} \leq 2\max|u|\,\|u - \tilde{u}\|^2_{L^\infty(\Omega)}.
\tag{69}
$$

$\square$

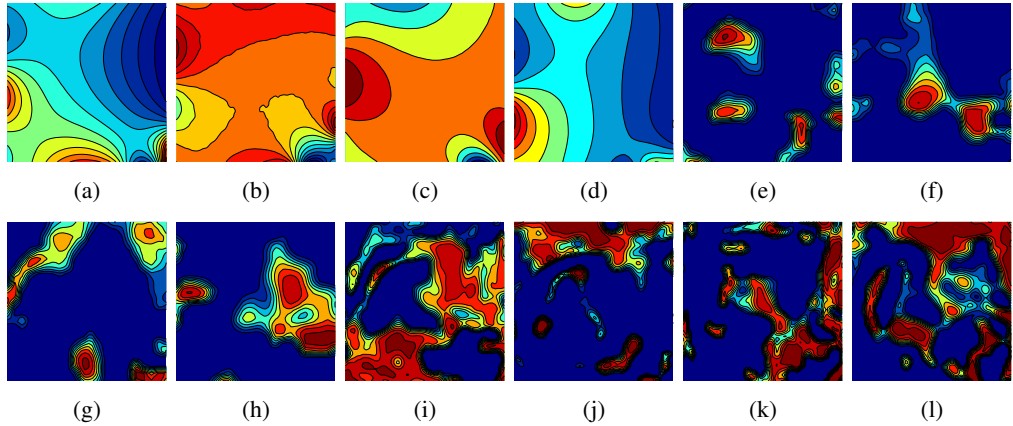

Figure 9: The latent representations from CNN-based UNet when evaluating for the sample in Figure 5, in their respective layers, 4 latent representations are extracted from 4 randomly selected channels: (a)–(d) are from the feature extracting layer (layer 1, $128 \times 128$ grid), (e)–(h) are from the middle layer acting on the coarsest level ($32 \times 32$ grid), (i)–(l) are from the next level ($64 \times 64$ grid).

Figure 10: The latent representations from UIT when evaluating for the sample in Figure 5, in their corresponding layers with respect to those in Figure 9, 4 latent representations are extracted from 4 randomly selected channels: (a)–(d) are from the feature extracting layer (layer 1, $128 \times 128$ grid), (e)–(h) are from the middle layer acting on the coarsest level ($32 \times 32$ grid), (i)–(l) are from the next level ($64 \times 64$ grid).

## G    LIMITATIONS, EXTENSIONS, AND FUTURE WORK

In this study, the $\sigma$ to be recovered relies on a piecewise constant assumption. This assumption is commonly seen in the theoretical study of the original DSM. For many EIT applications in medical imaging and industrial monitoring, $\sigma$ may involve non-sharp transitions or even contain highly anisotropic/multiscale behaviors making it merely an $L^\infty$ function. If the boundary data pairs are still quite limited, i.e., only a few electric modes are placed on the boundary $\partial\Omega$, the proposed model alone is not expected to perform as well as in benchmark problems. Nevertheless, it can still contribute to achieving reconstruction with satisfactory accuracy, if certain a priori knowledge of the problem is accessible. End2end-wise, our proposed method has limitations like other operator learners: the data manifold on which the operator is learned is assumed to exhibit low-dimensional/low-rank attributes. The behavior of the operator of interest on a compact subset is assumed to be reasonably well approximated by a finite number of bases. Therefore, for non-piecewise constant conductivities, the modification can be to employ a suitable data set, in which the sampling of $\{\sigma^{(k)}\}$ represents the true $\sigma$'s distribution a posteriori to a certain degree. However, to reconstruct non-piecewise constant conductivities, more boundary data pairs or even the entire NtD map is demanded from a theoretical perspective (Astala & Päivärinta, 2006; Nachman, 1996; Kohn & Vogelius, 1984; Sylvester & Uhlmann, 1987). For fewer data pairs and more complicated conductivity set-up, there have been efforts in this direction hierarchically using matrix completion (Bui-Thanh et al., 2022) to recover $\Lambda_\sigma$. When $\Lambda_\sigma$ is indeed available, $\sigma$ can be described by a Fredholm integral equation, see Nachman (1996, Theorem 4.1), which itself is strongly related to the modified attention mechanism (14) of the proposed Transformer. The architectural resemblance may lead to future explorations in this direction. Optimization with regularization can be applied for the instance of interest (fine-tune) from the perspective of improving the reconstruction for a single instance. This approach dates back to the classical iterative methods involving adaptivity (Jin & Xu, 2019). Recent novel DL-inspired adaptions (Li et al., 2021b; Benitez et al., 2023) re-introduce this type of method. In fine-tuning, the initial guess is the reconstruction by the operator learner trained in the end2end pipeline (pre-train).

