# OpenReview forum: "Transformer Meets Boundary Value Inverse Problems"
_ICLR.cc/2023/Conference — ICLR 2023 poster_

### Official Review · Reviewer_ne3Z · 2022-10-14

**Confidence:** 4
**Clarity, Quality, Novelty And Reproducibility:** The approach is novel and well presen…
**Correctness:** 3
**Technical Novelty And Significance:** 3
**Empirical Novelty And Significance:** 3
**Recommendation:** 8

**Strength And Weaknesses:**

Strengths:
I feel that the paper has potential as it proposes a new concept to solve the boundary value inverse problems (at least in theory). They combine attention mechanism from transformers to solution and provide theoretical background/justification for that why it is done. This is interesting new approach. The paper is also written relatively well and derivations are understandable.

Weaknesses:
I feel that the study is quite theoretical as it considers a very simplified setup: the unknown is assumed to be an inclusion with known conductivity with also known background conductivity. This setup is quite commonly used assumption in more theoretical studies related to EIT and therefore would not high weight on this. But I would still like to also see some discussion about possible extensions for more practical setups. For example, how the approach could be extended to conductivity distributions which have non-sharp transitions. There are several examples of such cases in medical imaging and monitoring of industrial processes.



**Summary Of The Paper:**

The paper proposes a new Transformer based architecture to certain inverse problems in which attempt is to reconstruct the structure of inner domain from boundary measurements. The problems are important as several practical applications fall into this class (e.g. EIT, optical tomography, seismic tomography). The study attempts to answer two questions: 1) how the boundary data should be fed to the model, and 2) what is structure of the neural network




**Summary Of The Review:**

Overall, I am positive about this study and feel that it deserves publications. But I would have some discussion about practical e.g. in future directions.

(Note: I have also previously reviewed this paper for another conference.)

---

> ### Author Response · Authors · 2022-11-18
> **Response to Reviewer ne3Z**
>
> We would like to express our appreciation for reviewer ne3Z's reviewing effort to help us to improve the paper one more time.
>
> For the question about a more practical setup (also from your question in a previous conference), after the short response period of that conference, realizing our response to this question was unsatisfactory, we brought this question to some of our colleagues working on traditional methods for some broader aspects in PDE-related inverse problems.
>
> So, here is our updated answer:
>
> - For inclusions with non-sharp transitions or even those that can be merely described by $L^{\infty}$ functions (which is not even Hilbert), the inverse problem itself will become much more challenging due to more degrees of freedom, e.g., not just the shapes and locations of, but also the changing values of $\sigma$. In this case, one usually can not expect too much accuracy if only a few boundary data pairs are available. A reasonable approach to get more accurate reconstructions would be to come up with a way to get an accurate approximation to the NtD mapping $\Lambda_{\sigma}$. However, even with limited boundary data pairs, by employing a suitable data set to capture the non-sharp transitions, which changes an ill-posed inverse problem to an interpolation problem in the high-dimensional data manifold (this point is also updated in the rebuttal revision in introduction), we believe the proposed pipeline can still result in reconstructions of satisfactory accuracy.
> - In the case of full measurement, there is a very interesting relation between the attention architecture and the Fredholm integral equation which describes $\sigma$ with the NtD mapping $\Lambda_{\sigma}$.
> - Using PDE-generated data with known inclusion conductivity with also known background conductivity gives us a verifiable ground truth for the forward problem (NtD maps), and this practice is usefully to do prototyping/proof of concept. With random medium and/or unknown conductivities, even the numerical methods for forward problem using standard discretizations (FEM/FDM on Cartesian grids) have troubles to approximate the solutions. This hinders the verifiability in practice of using these data to develop DL models further. In this synthetic setup, at least we know confidently the difference between what we tried to approximate and what the DL models yield, and can work on that.
> - After the end2end operator learning stage, the other way to deal with more complicated inclusion would be an instance-based optimization approach such as Tikhonov regularization, while using the evaluation result of the operator learner as the initial guess. For the forward PDE problem, this was explored in [3], which mimics the pretrain-finetune pipeline in the CV and NLP community.
>
> All the topics above could be potentially interesting to explore by the joined community of IP and DL in the future. Inspired by the reviewer's comment, we have specifically added one section in Appendix (Appendix G) to discuss these possible future research possibilities.
>
> [3]: Li, Z., Zheng, H., Kovachki, N., Jin, D., Chen, H., Liu, B., ... & Anandkumar, A. (2021). Physics-informed neural operator for learning partial differential equations. arXiv preprint arXiv:2111.03794.

---

> > ### Comment · Reviewer_ne3Z · 2022-11-22
> > **Reply**
> >
> > I am happy to see added appendix for limitations and future work. For "we believe the proposed pipeline can still result in reconstructions of satisfactory accuracy", I am having some doubts as currently there is only one inclusion considered and often is hardly enough to approximate the conductivity distribution. But extensions for more than one inclusion?

---

> > > ### Author Response · Authors · 2022-11-23
> > > **Further Response to Reviewer ne3Z part 2**
> > >
> > > Not 100% sure what "one inclusion" is referred to here. But, based on our understanding and the methodology described above, we offer our thoughts as below.
> > > - If you mean the case of non-sharp inclusion boundary, then in order to apply the current pipeline to this case, we need to generate a suitable training data set that can capture behavior of non-sharp transitions. For example, an ellipse in our current training data set follows $(x-x_0)^2/a^2+(y-y_0)^2/b^2=1$, where $x_0, y_0, a, b$ are random. If we are still interested in (nearly)elliptical-shaped inclusions, for non-sharp transitions, the equation above may be replaced by $\arctan(((x-x_0)^2/a^2+(y-y_0)^2/b^2-1)/\epsilon)/\pi + c$, and $a,b,c, \epsilon, x_0, y_0$, and orientation are sampled from distributions. Knowning $\epsilon$ relatively big ($>1$), this would generate a dataset with non-sharp transition. If a priori the diffusive interfaces are assumed to be captured by the transition layer of $\arctan$, then (a) we can exploit the flexibility of DNNs and use $\arctan$ as the last layer's activation and let the autograd take over; (b) such a "compactly" sampled dataset can be reasonably well-learned by the Transformer architecture in general.
> > > - If you mean that the inclusion(s) follows a distribution, instead of being a "fixed" configuration for a single instance, then it is possible to use Transformer-like architecture to do Bayesian-type inference to approximate a distribution [4]. Of course, once entering the Bayesian realm, the topic widens by several orders of magnitude, and one may want to reformulate the problem as a Bayesian inverse problem. In this paper, we would like to stick to the operator learning framework introduced by the FNO paper [5]: use operator learning to "interpolate" or "extrapolate" a compactly sampled dataset.
> > > - However, things might become more and more problematic if the submanifold on which the data are sampled grows less and less compact (relative to the overall $L^2$ or $L^{\infty}$), e.g., it contains much multiscale oscillations. The proposed pipeline may be too toy-model'ish, and we may need to explore a more problem-oriented network to tackle a complicated distribution, such as material having multiscale structures. In these cases, much more spectral information from the NtD mapping may be needed. Then, again, if one can convert the ill-posed problem to a "well-posed" data interpolation problem in certain low-dimensional data manifold in a high dimensional space, the operator learning framework from [5] and [6] would work again. Yet how much more data are needed in that one can trade the missing spectral information with these sampling will be an interesting topic to study.
> > >
> > > [4]: Müller, Samuel, et al. "Transformers Can Do Bayesian Inference." *ICLR 2022*.
> > >
> > > [5]: Li, Zongyi, et al. "Fourier neural operator for parametric partial differential equations." *ICLR 2021*.
> > >
> > > [6]: N. H. Nelsen, A. M. Stuart; The Random Feature Model for Input-Output Maps between Banach Spaces, SIAM J. Sci. Comput., 43(5), A3212–A3243.
> > >
> > > > But extensions for more than one inclusion?
> > >
> > > If the reviewer means multiple inclusions with different conductivity values, then the "the dataset" will be the savior once again, i.e., the inclusion of interest ought to lie within the support of this compact sampling. Yet, in Section 3.2, 3.4, the presented learnable direct sampling itself does not need the assumption of a single conductivity value for inclusions.

---

> > > ### Author Response · Authors · 2022-11-23
> > > **Further Response to Reviewer ne3Z part 1**
> > >
> > > > For "we believe the proposed pipeline can still result in reconstructions of satisfactory accuracy", I am having some doubts as currently there is only one inclusion considered and often is hardly enough to approximate the conductivity distribution.
> > >
> > > The base of our expectation comes from the reformulation of the ill-posedness of inverse problems. Let's briefly recall the methodology. The operators
> > >
> > > $$
> > > \mathcal{F}: \sigma \mapsto \Lambda_{\sigma}, \quad  \text{ and } \quad  \mathcal{F}^{-1}: \Lambda_{\sigma} \mapsto \sigma,
> > > $$
> > >
> > > are well defined.  Consider the operators with finite data pair
> > >
> > > $$
> > > \mathcal{F}_L: \sigma \mapsto \\{ (g_1, \Lambda\_{\sigma} g_1), \dots, (g_L, \Lambda\_{\sigma} g_L) \\} \quad \text{ and } \quad \mathcal{F}^{-1}_L: \\{ (g_1, \Lambda\_{\sigma} g_1), \dots, (g_L, \Lambda\_{\sigma} g_L) \\} \mapsto \sigma.
> > > $$
> > >
> > > The inverse operator $\mathcal{F}^{-1}_L$ is ill-posed.  Instead of approximating the ill-posed $\mathcal{F}^{-1}_L$, the proposed operator learning approach aims to "learn"
> > >
> > > $$
> > > \mathcal{F}^{-1}\_{L,\mathbb{D}}: \\{ (g_1, \Lambda\_{\sigma^{(k)}} g_1), ..., (g_L, \Lambda\_{\sigma^{(k)}} g_L) \\} \mapsto \sigma^{(k)}, \quad k= 1,\dots,N.
> > > $$
> > >
> > > where  $\mathbb{D}:=\{ \sigma^{(k)}\}\_{k=1}^N$ is a compactly sampled data set. The incomplete information of $\Lambda_{\sigma}$ due to a small $L$ for one single $\sigma$ is compensated by a large $N\gg 1$ sampling of different $\sigma$'s.  Then the ill-posed problem is tamed to be become a (relatively) well-defined data interpolation problem. The coefficient $\sigma$ is not necessarily to be a piecewise constant function on one connected sub-domain $D$. The key is to generate large data $\{ (g_1, \Lambda_{\sigma^{(k)}} g_1), ..., (g_L, \Lambda_{\sigma^{(k)}} g_L) \}, k= 1,\dots,N\gg 1$ s.t. these data are separable and the true coefficient function $\sigma^{(k)}$ serves as the role of label (a 2D imagine label not just a scalar label). We do believe these data is separable if we sample  $\sigma^{(k)}$ following certain distribution which can be considered as sampling on a lower dimensional sub-mainfold of the space $L^2(\Omega)$ or $L^{\infty}(\Omega)$. We call it a "compactly" sampled dataset. So our problem now becomes how to better interpolate these data.

---

### Official Review · Reviewer_CBcr · 2022-10-22

**Confidence:** 3
**Correctness:** 3
**Technical Novelty And Significance:** 3
**Empirical Novelty And Significance:** 3
**Recommendation:** 5

**Clarity, Quality, Novelty And Reproducibility:**

The paper clarity is not satisfactory. The datasets and codes are suggested to be released for reproducibility.

**Strength And Weaknesses:**

1. Strength

(1) The paper tackles an important inverse problem (boundary value problem) by designing attention-based neural network architecture.

(2) The proposed attention-like learnable operators have novelty and mathematical foundation.

(3) The proposed network for inverse operator of electrical impedance tomography shows good results.

2. Weakness

The major weakness  this paper are on the evaluation of effectiveness of the proposed  transformer, the unclear presentation of the proposed method, and the missing training and experimental details and comparisons.

(1)  The paper designed the attention mechanism in Eqn. (9) deriving the non-local kernel with theoretical foundation in Theorem 2. In the section 3.4, the index map integral is also formulated as attention. In these subsections, the modeling of ingredients in electrical impedance tomography are formulated as attentions. What are the advantages of the proposed  attention? These attention might be also possible to further extended with learnable parameters, might further increase the learning power. It is also unclear to me, how these proposed attentions construct a deep network? The network training loss and details should be also presented in the major part of the paper.

(2) In the experiments, the proposed method is compared with the CNN-based U-Net, FNO, and also variants of U-Net with different attentions. The proposed attention shows better performance. However, in the experimental sections, their is no detailed introduction to the experimental datasets, training details, network parameter sizes, etc.  The comparisons with more neural operator learning methods (refer to the related works) should be also given.

(3) Overall, the organization of the paper can be improved. It is hard to read to me in the current version, mainly because of missing explanations to the global network architecture, the rule of the proposed modulus in the network architecture, and training/experimental details.

**Summary Of The Paper:**

This paper works on the solving boundary value inverse problem by designing the transformer network layers inspired by the mathematics foundations in the boundary value problem. Specifically, it proposed attention-based transformer by learnable non-local kernel, with application to electrical impedance tomography.  The proposed network architecture is based on learning inverse operator by a harmonic extension and an integral operator with the non-local attention kernel. Experiments show better results than the compared U-Nets and inverse operator, including FNO.

**Summary Of The Review:**

The good point of this paper is the proposed attention modules for designing deep network for learning inverse operators, with better performance than the FNO and other baselines with U-net as backbone. However, the paper is hard to read, and has limitations in experiments. Please refer to the weakness for details.

---

> ### Author Response · Authors · 2022-11-18
> **Response to Reviewer CBcr part 2**
>
> > how these proposed attentions construct a deep network? The network training loss and details should be also presented in the major part of the paper.
>
> Due to the page limitation, we have originally presented these in Appendix B. The training loss is now equation (8). Now much more has been added or updated (Figure 1, Figure 7, Figure 8, Table 3, and Appendix C).
>
>
> > The comparisons with more neural operator learning methods (refer to the related works) should be also given.
>
> We added two operator learning models from ICLR 2022: MultiWavelet Neural Operator (MWO) from [https://openreview.net/forum?id=d2TT6gK9qZn](https://openreview.net/forum?id=d2TT6gK9qZn) and a modified FNO adding the adaptive token mixing from [https://openreview.net/forum?id=EXHG-A3jlM](https://openreview.net/forum?id=EXHG-A3jlM) reviewer 5fLN mentioned. The corresponding source codes have been updated in `mwo.py` and `fno.py` in the supplemental mateiral, together with their configurations updated in `configs.yml`. The setup of MWO is taken from the official repo that is used to infer a 2D forward PDE [https://github.com/gaurav71531/mwt-operator/blob/master/Darcy.ipynb](https://github.com/gaurav71531/mwt-operator/blob/master/Darcy.ipynb).
>
> > Overall, the organization of the paper can be improved. It is hard to read to me in the current version, mainly because of missing explanations to the global network architecture, the rule of the proposed modulus in the network architecture, and training/experimental details.
>
> Inspired by this comment, as well as relevant ones from the other reviewers, we did a major reorganization of the paper to try our best to make the paper more clear and accessible. Here, we have specifically made the following revisions which, we hope, can address the reviewer's concerns:
>
> - We have added description about data generalization at the beginning of Section 3.1, Section 4, and details in Appendix C.1.
> - We have updated the training in Section 4, and much more details about the training, network architecture comparisons in Appendix C.
> - We have added the benchmark results for two neural operators featured in ICLR 2022: adaptive FNO2d with several token mixing layers in frequency, and MultiWavelet Neural Operator.
> - Plots of latent representations (Figures 9 and 10) are added in the Appendix to show empirically how Theorem 2 works in action.

---

> ### Author Response · Authors · 2022-11-19
> **Response to Reviewer CBcr part 1**
>
> ### Summary
>
> > The paper clarity is not satisfactory.
>
> We greatly appreciated reviewer CBcr for the time and effort to help us to improve the paper. We have realized this (hopefully not too late) after studying the review carefully. In the rebuttal revision, we have put great effort to explain and clarify things up. We invite you to read the paper again to see if the clarify has been improved. Major revision to address reviewer CBcr's questions include:
>
> - The introduction and Section 3 have largely overhauled to lay out our story following the thread of thoughts below:
>     - boundary value inverse problem is hard (the operator itself is ill-posed) &rArr;
>     - so we use operator learning to tame the ill-posedness &rArr;
>     - but how to find suitable data format? (harmonic extension fits the tensor2tensor pipeline) &rArr;
>     - how about the basic block in the network? (kernel integration as attention) &rArr;
>     - why it may be good for this problem? (DSM and attention have similar forms and have global dependence).
> - More neural operators from ICLR 2022 have been compared, experiment details and network architecture comparison have been added/updated in Appendix C.
>
> > The datasets and codes are suggested to be released for reproducibility.
>
> The codes are in the supplemental material, the training of every model uses the same setup (Appendix C, see also the `configs.yml` file). The dataset, as well as the trained model using the configurations in `configs.yml` file, is available for download at [https://www.kaggle.com/datasets/anonymousauthor25/eit-transformer](https://www.kaggle.com/datasets/anonymousauthor25/eit-transformer).
>
> &nbsp;
> ### Answers to individual questions/concerns
>
> > What are the advantages of the proposed attention?
>
> Rather than proposing a new type of neural operator and applying it to every forward PDE benchmark out there, we carefully design a benchmark based on practical interest (boundary value inverse problem) to let attention's capacity to model long range interaction (but without smoothing out eigenmodes) shine:
>
> - Its architecture is conforming to the mathematical structure of the direct sampling method, and we believe it is one of keys for the nice reconstruction results.
>
> - It is able to, both in theoretical capacity and in empirical evidence, recover high-frequency information by bootstrapping low-frequency data. Based on Theorem 1 and limited data pair constraint, we capitalize attention's frequency-bootstrapping capacity on spectral information of the NtD operator (use attention to "bootstrap" higher modes from lower modes). Learning more information higher up in the spectrum from only lower modes definitely helps the reconstruction. Some empirical evidence has been added in the appendix on page 28.
>
> > These attentions might be also possible to further extended with learnable parameters, might further increase the learning power.
>
> The three learnable matrices $W^Q$, $W^K$ and $W^V$ can come directly from the original mathematical structure of DSM (Section 3.4, see also Appendix D). We agree with the reviewer that the extension with other learnable structures is possible. We chose DSM as it conforms with attention a lot layerwise: probing direction is like query, based on how it responds with the feature maps (PDE-based harmonic extensions as keys), the value delivers the reconstruction. However, as of now, human knowledge does not yet know the mysterious mathematical structure for the inverse operator for EIT with only finite measurements, further extending the attentions suitably such that it maintains the conformity to the mathematical structure is unknown to us.

---

### Official Review · Reviewer_5fLN · 2022-10-23

**Confidence:** 3
**Correctness:** 3
**Technical Novelty And Significance:** 2
**Empirical Novelty And Significance:** 3
**Recommendation:** 8

**Clarity, Quality, Novelty And Reproducibility:**

The clarity of the paper (especially Section 3.1) can be improved in a few ways -- see my comments above. The quality and novelty of the work are high in my opinion. The proposed operator learning model, described in detail in Appendix B.2, seems reproducible.


**Strength And Weaknesses:**

# Strength
The proposed transformer-based operator model is novel and achieves strong performance. The "frequency bootstrapping" property demonstrated for the attention mechanism in Theorem 2 is revealing -- It may be of independent interest to general applications of transformers beyond inverse problems.

# Weakness
I was confused by the writing of the authors in quite a few places. Below I list a few of them.

- Is the sum $\sum_{m \in \mathbb{M}_L}$ in Equation (2) extraneous? First, the letter $m$ in the sum is not defined. Second, if $m$ is meant to be $\mathbf{m}$, then one should probably write an integral instead of a sum -- the space $\mathbb{M}_L$ is not discrete in general. Last, even if the integral is used to replace the sum,  I still do not see why one wants to find a ground-truth signal $\mathbf{p}$ so that its forward process is close to all **all** possible measurements $\mathbf{m} \in \mathbb{M}$. The curly bracket in Equation (2) was also placed in a confusing way -- I assume that the regularization term should be included in $\inf$.

- What confuses me the most is Section 3.1. There seem to be a couple of glitches in the current formulation. At the bottom of page 3, the authors wrote "Then, the coefficient of (4) to be recovered can be described by a characteristic function $\mathbf{p} = \mathcal{I}^D(x)$ defined for ..." Later, in the same sentence at the beginning of page 4, the authors wrote "or equivalently, $\mathbf{p}=\sigma_1 \mathcal{I}^D+\sigma_0\left(1-\mathcal{I}^D\right)$." The definition of $\mathbf{p}$ at these two different places is inconsistent.

- If I understand it correctly, the object $\mathbf{p}=\sigma_1 \mathcal{I}^D+\sigma_0\left(1-\mathcal{I}^D\right)$ is what we aim to recover in EIT.  However, in the paragraph before the paragraph of Equation (6), it was mentioned that $\mathcal{F}^{-1}: \mathbb{M} \to \mathbb{P}$ is "essentially a map $\Lambda_{\sigma} \mapsto \sigma$".  I am baffled by this sentence. Indeed, $\Lambda_{\sigma}$ is not an element of $\mathbb{M}$, and $\sigma$ is not an element of $\mathbb{P}$. In addition, I thought that $\sigma$ is known -- in lines after Equation (4), the authors mentioned "The values $\sigma_0$ and $\sigma_1$ are two (approximately) known constants...". So my understanding is that the unknown in  $\mathbf{p}=\sigma_1 \mathcal{I}^D+\sigma_0\left(1-\mathcal{I}^D\right)$ is $D$, not $\sigma$. Could the authors correct me if this is wrong? If $\sigma$ is the unknown, then the authors may want to remove the confusing sentence "The values $\sigma_0$ and $\sigma_1$ are two (approximately) known constants...".  If $D$ is the unknown, how does one recover $\mathbf{p}=\sigma_1 \mathcal{I}^D+\sigma_0\left(1-\mathcal{I}^D\right)$ from $\sigma = (\sigma_0, \sigma_1)$ yielded from $\Lambda_{\sigma} \mapsto \sigma$, when $D$ is not given?

- In general, to improve clarity in Section 3.1, I think it would be great if the authors could explicitly relate the forward operator in EIT to Equation (1) -- what do $\mathbf{m}$, $\mathbf{p}$, and $\mathcal{F}$ stands for in the EIT problem.

- In section 3.2, the authors cast the attention mechanism as a kernel integral operator. A similar observation has been previously shown in [1]

- In the experiment section, it would be great if the authors could comment/demonstrate the mesh-invariance of the proposed operator model. For instance, the authors could train their operator model on one mesh discretization and evaluate it on another discretization ("zero-shot superresolution"), and see if the performance degrades substantially. Similar experiments have been reported by many earlier neural operator papers -- the authors can use them as benchmarks.

- The experiment section focuses on an EIT problem. Have the authors considered other generic operator-learning problems, e.g., generic parametric PDE datasets? Since EIT is the focus of this paper, it would be unfair to ask the authors to train their models on these irrelevant datasets -- but I would be curious to learn if the authors have tried before.

- Typo: In "2.1 Contributions", second bullet point, Transformeris -> Transformers

[1] Guibas J, Mardani M, Li Z, Tao A, Anandkumar A, Catanzaro B. Adaptive Fourier neural operators: Efficient token mixers for transformers. ICLR 2022




**Summary Of The Paper:**

This paper proposes to use the transformer architecture to solve boundary value inverse problems.

Theoretically, the authors justify the desirable properties of Transformers in boundary-value inverse problems in two ways: (i) an attention-based mechanism can generate an output signal with high frequencies than its input (Theorem 2), and (ii) the index map integral of boundary-value problems is similar to an attention mechanism in form (Equation (25)).

Empirically, the authors demonstrated the strong performance of transformer-based architecture in an electrical impedance tomography problem.

**Summary Of The Review:**

I feel that the clarity of the paper (especially Section 3.1) can be improved in a few ways. As mentioned above, I think that the current formulation can be a bit confusing; there may exist a few glitches. Other than that, I think the work is solid.

---

> ### Author Response · Authors · 2022-11-18
> **Response to Reviewer 5fLN part 3**
>
> > In section 3.2, the authors cast the attention mechanism as a kernel integral operator. A similar observation has been previously shown in [1].
> >
> > [1] Guibas J et al. Adaptive Fourier neural operators: Efficient token mixers for transformers. ICLR 2022.
>
>  We are grateful for the reviewer to let us know this important reference. We have added in the rebuttal revision. We also would to like to explain more of our contribution of our work, different from that in [1], from the following two aspects.
>
>  - First and foremost is to connect DSM of EIT and the attention through a kernel integral operator, which is not trivial and not done before due to complexity of the EIT itself.
>
>  - We would like to emphasize that, the kernel here is **instance-dependent** during forward pass. Originally the Pincherle-Goursat kernel argument was hidden in the proof of the "frequency bootstrapping" theorem. Now in the rebuttal revision, this "instance-dependent" nature of the attention-like integral kernel is emphasized in the main body. To explain this "instance-dependency", let us revisit CNN (or FNO): after being trained, CNN's (or FNO's) filter is data-independent in evaluation's forward pass. Let us take the kernel $\kappa_{\theta}(x-y)$ in CNN (or $\mathrm{FFT}(\kappa_{\theta}(x,\cdot))$ in FNO) for example, these filters do not depend on an instance during forward pass. This is to say, if the feature itself in this instance does not have any high frequency information, even if the trained filter is able to capture high frequency features (e.g., Sobel filter to extract edges), it cannot extract these feature from this pure low frequency mode. In contrast, attention's kernel $\kappa_{\theta}(x,y;\phi)$ is instance-dependent even during evaluation and can attribute to a more instance-dependent "frequency bootstrapping" layerwise speaking.
>
> > In the experiment section, it would be great if the authors could comment/demonstrate the mesh-invariance of the proposed operator model.
>
> Unfortunately the current model's evaluation is not mesh-size-invariant, and actually none of operator learners we have tested are for this EIT inverse problem. Even for Fourier Neural Operator, the best overall mesh-invariant operator learner for forward PDE problems, is not mesh size invariant for this EIT benchmark. For FNO2d, even trained on a higher resolution (201x201), the predictions totally breaks down at a lower resolution during evaluation (101x101), following the specialized positional encoding setup of "zero-shot super-resolution". Note that FNO2d can do the reverse ("zero-shot super-resolution" which is more impressive) for the forward PDE problems (trained at a lower resolution, evaluation at a higher resolution). For forward PDE problems (e.g., the Burgers', Darcy flow, and Navier-Stokes benchmarks) in the earlier neural operator papers, all of them have either (1) spectral decaying property: e.g., Burgers' and NS, as the input data is generated by Gaussian Random Field, the viscous term in Burgers' and the elliptic nature of the vorticity formulation (the NS formulation the original FNO paper opts) smoothen out the higher modes so the operator's behavior can be captured by lower modes; (2) low-rank nature: e.g., Darcy's flow example, please check Figure 3 in [2] where in certain latent space, the operator we want to learn is a nonlinear perturbation to the identity. For inverse problem, we do not have these nice mathematical structures as the forward PDE model has. We added a mesh-size-invariant test for FNO2d in the source code in the supplemental material.
>
> [2]: N. H. Nelsen, A. M. Stuart; The Random Feature Model for Input-Output Maps between Banach Spaces, SIAM J. Sci. Comput., 43(5), A3212–A3243.
>
> > The experiment section focuses on an EIT problem. Have the authors considered other generic operator-learning problems, e.g., generic parametric PDE datasets?.
>
> We appreciate the reviewer bringing this up. Our approach here is rather "designing a novel and interesting benchmark problem that the new architecture excels because it is more structure-conforming", than "applying a new model to all the existing benchmarks and see how it goes". Our honest answer to this question is: when prototyping this paper earlier this year, we tried the operator learner on the Darcy's flow benchmark other neural operator papers used, and we tried and tweaked various architectures to surpass the best operator learner FNO's performance (by that time) just to make sure it works first for a 2D forward PDE problem, otherwise we would stop there back then.
>
> > Typo: In "2.1 Contributions", second bullet point, Transformeris -> Transformers.
>
>  Thanks, we have corrected it.

---

> ### Author Response · Authors · 2022-11-18
> **Response to Reviewer 5fLN part 2**
>
> >  What confuses me the most is Section 3.1. There seem to be a couple of glitches in the current formulation. If I understand it correctly, the object $\mathbf{p} = \sigma_1\mathcal{I}^D + \sigma_0(1-\mathcal{I}^D)$ is what we aim to recover in EIT. However, in the paragraph before the paragraph of Equation (6), it was mentioned that $\mathcal{F}^{-1}: \mathbb{M} \rightarrow \mathbb{P}$ is `essentially a map $\Lambda_{\sigma}\mapsto\sigma$". I am baffled by this sentence. Indeed, $\Lambda_{\sigma}$ is not an element of $\mathbb{M}$, and $\sigma$ is not an element of $\mathbb{P}$.
>
> > In addition, I thought that $\sigma$ is known -- in lines after Equation (4), the authors mentioned "The values $\sigma_0$ and $\sigma_1$ are two (approximately) known constants...". So my understanding is that the unknown in $\mathbf{p} = \sigma_1\mathcal{I}^D + \sigma_0(1-\mathcal{I}^D)$ is $D$ not $\sigma$. Could the authors correct me if this is wrong?  If $\sigma$ is the unknown, then the authors may want to remove the confusing sentence "The values $\sigma_0$ and $\sigma_1$ are two (approximately) known constants...". If $D$ is the unknown, how does one recover $\mathbf{p} = \sigma_1\mathcal{I}^D + \sigma_0(1-\mathcal{I}^D)$ from $\sigma = (\sigma_0,\sigma_1)$ yielded from $\Lambda_{\sigma}\rightarrow \sigma$ when $D$ is not given.
>
> Thanks for this comment, we also found the definition of $\sigma$ is not clear. In EIT, $\sigma$  is considered as a piecewise constant function, while $\sigma_0$ and $\sigma_1$ are two constants representing the value of $\sigma$ outside and inside the inclusion $D$, respectively. The values taken, i.e.,  $\sigma_0$ and $\sigma_1$ are known. What is unknown is the shape/location of the discontinuity, i.e., the boundary of the inclusion $D$. The index function $\mathcal{I}^D$ (the characteristic function of $D$) is equivalent to the knowledge of $\sigma = \sigma_1\mathcal{I}^D + \sigma_0(1-\mathcal{I}^D)$. So both $\sigma$ and $D$ are unknown and indeed they are equivalent through the introduction of $\mathcal I^D$.
>
> Now we define $\sigma$ as a piecewise constant function right after PDE (1). The beginning of Section 3.1 (now at the beginning of Section 3.2) has also been rewritten to hopefully clarify this confusion.
>
> We also realized our notation $\mathbb M$ and $\mathbb P$ on the spaces are not precise. Our original presentation aims to treat an abstract inverse problem, and as a result it was too ambitious. Now we have removed $\mathbb M$ and $\mathbb P$ and focus on EIT only. Define the forward operator and inverse operator
>
> $$
> \mathcal{F}: \sigma \mapsto \Lambda\_{\sigma}, \quad \text{ and } \quad \mathcal{F}^{-1}: \Lambda\_{\sigma} \mapsto \sigma,
> $$
> where $\sigma$ is a piecewise constant function and $\Lambda_{\sigma}$ is the NtD mapping which can be represented by an infinite dimension matrix $\mathbf{A}_{\sigma}$. Both $\mathcal F$ and $\mathcal F^{-1}$ are well defined.
>
> If we know infinitely many boundary data pairs $(g_l,f_l=\Lambda\_{\sigma}g_l)$, $l=1,2,...$, , we can use them to determine $\Lambda_{\sigma}$ and then $\sigma = \mathcal F^{-1}(\Lambda\_\sigma)$. From the perspective of spectral theory, "knowing" all the boundary data pairs (functions) is equivalent to "knowing" $\Lambda\_{\sigma}$ (an operator map). That was why we say it is essentially "$\Lambda\_{\sigma}\mapsto\sigma$".
>
> We brought this setup from theoretical studies of EIT without giving it too much attention. Now, more details about EIT are added in Appendix B, and we wrote one analytical example to show how this is done in theory and where the difficulty may lie.
>
> Different from theoretical studies of EIT, what we are interested in this work is to use only finite/a few data pairs $(g_l,f_l=\Lambda_{\sigma}g_l)$, $l=1,2,...,L$ for reconstruction. Now we have realized it was confusing, and we have completely rewritten the introduction to make our problem of interest more clear and more specific.
>
> > In general, to improve clarity in Section 3.1, I think it would be great if the authors could explicitly relate the forward operator in EIT to Equation (1) -- what do $\mathbf{m}$, $\mathbf{p}$, and $\mathcal{F}$  stand for in the EIT problem.
>
> Inspired by reviewer 5fLN but also all other reviewers' reviews, in hindsight we should not be too ambitious to use general notations to cover all boundary value inverse problems. In the current rebuttal revision, we have switched to a more EIT-dedicated presentation in the introduction, we invite you to read Section 1 again to see if things get more clear. In the new presentation, we spilled out explicitly the meaning of the forward and inverse operators as well as their input and outputs at the beginning of the introduction.

---

> ### Author Response · Authors · 2022-11-18
> **Response to Reviewer 5fLN part 1**
>
> ### Summary
> We greatly appreciate the time and effort reviewer 5fLN put to help us better the presentation of our paper. The major weakness in the original submission was:
>
> > The clarity of the paper (especially Section 3.1) can be improved in a few ways -- see my comments above.
>
> Based on the reviewer 5fLN as well as other reviewers' comments, we realized that our original presentation was too ambitious, takes too much for granted, and is not very audience-friendly. In the rebuttal revision, we have changed a lot in terms of presenting the problem and our thoughts. We sincerely invite reviewer 5fLN to read the paper again to see if it clarifies your question a bit in the review.
>
>  - Instead of presenting a general formulation for inverse problems using $\mathbf{m}$, $\mathbb{M}$, etc., we have now completely rewritten the introduction using the EIT problem's notation.
>
>  - Sections 3.2 has been revised a lot to (1) clarify the notations ($\sigma$, NtD maps and data pairs, etc.); (2) explain the motivation of using $\nabla\phi$ as the inputs. Section 3.4 focuses now on more of the rationale behind using DSM to develop attention-like architectures, and a more detailed explanation of technicality is moved to Appendix D.
>
>  - The reference reviewer 5fLN mentioned is added.
>
>  - More explanation of how attention is an instance-base kernel is added in Section 3.3 (page 7) as well as in the Appendix on page 25.
>  - A detailed review and explanation of the theoretical background of EIT is added in Appendix B, including which targets to be recovered and which data is used as well as the uniqueness result. An explicit example is provided to help readers from various backgrounds better understand this problem.
>
> &nbsp;
>
> ### Detailed answer to each question
>
> > Is the sum $\sum_{m\in\mathbb{M}_L}$ in Equation (2) extraneous? First, the letter $m$ in the sum is not defined. Second, if $m$ is meant to be $\mathbf{m}$, then one should probably write an integral instead of a sum -- the space $\mathbb{M}_L$ is not discrete in general. Last, even if the integral is used to replace the sum, I still do not see why one wants to find a ground-truth signal $\mathbf{p}$ so that its forward process is close to all possible measurements $m\in\mathbb{M}$.
>
>  In the original submission, we intended to present a general framework for inverse problems. However, this effort caused quite a few confusions and some misuse of notation. In the rebuttal revision, inspired by your review (as well as 4qwm's), we focus the presentation only on EIT to hopefully improve the clarity a bit. Equation (2) ((7) in the rebuttal revision) has also been corrected accordingly. Speaking of the original notation $\mathbf{p}$ and $\mathbf{m}$, they are exactly $\sigma$ and $(g,f=\Lambda_{\sigma}g)$, respectively, in the current rebuttal revision.
>
> The reconstruction strategy of EIT involves applying various currents $g$ on the boundary and measure the resulting voltages $f$. As a result, there are multiple boundary data pairs at hand, which contributes to the effort of recovering the ground truth of $\sigma$ ($\mathbf{p}$ in the original submission).  The sum is needed since ideally $\sigma$ can be only recovered by infinite boundary data pairs.
>
> We have updated Figure 1 to hopefully be more illustrative of this process. We have also added Appendix B to give an example-based review of the background of EIT.
>
> > The curly bracket in Equation (2) was also placed in a confusing way -- I assume that the regularization term should be included in inf.
>
> We thank the reviewer pointing out this discrepancy, and it is now corrected as equation (7).

---

> ### Comment · Reviewer_5fLN · 2022-11-21
> **Thanks for your responses**
>
> I'd like to thank the authors for their detailed responses and the significant text improvement in the new submission. They have addressed many of my concerns. I have raised my score accordingly.

---

### Official Review · Reviewer_4qwm · 2022-10-29

**Confidence:** 3
**Correctness:** 3
**Technical Novelty And Significance:** 3
**Empirical Novelty And Significance:** 3
**Recommendation:** 8

**Clarity, Quality, Novelty And Reproducibility:**

I find the paper quite unclear. As far as I can tell the work is novel and reproducible.

**Strength And Weaknesses:**

The authors design a transformer-based architecture to solve EIT (and more generally boundary value inverse problems). Their design is inspired by the mathematical / computational structure of the problem which is in a certain sense analogous to that of the attention layer. This is related to but different from physics-driven deep learning where usually the forward operator is somehow embedded in the network design. The present contribution is rather about shaping the inductive bias of the network for a specific class of problems.

I very much like that the authors address the problem by looking at operator-valued data (namely, the NtD map) and its discretization. This is the "right way" in the inverse problems community but earlier approaches mostly employ ad hoc solutions. Another nice thing in the manuscript is a principled proposal to "continue" the boundary data into the interior via harmonic extensions. These extensions can then be used as inputs to the transformer network. Numerical results show that their proposed method outperforms several strong baselines.

On the critical side, I find the paper very hard to follow. The prose could use a great deal of work---there are numerous misprints, non-idiomatic constructions, and broken and confusing sentences. Since this is very technical and specialist material that cannot be easily picked up "on the go", this will present a huge challenge to most ICLR readers. Just as importantly, many explanations are non-intuitive and  require background in EIT to follow. This is unfortunate because a typical ICLR reader will not have this background even if they are familiar with inverse problems in a different context. For comparison, earlier operator-learning papers like the FNOs are much easier to follow.

I'll give some examples to support my claims but they are far from exhaustive:

- "... the problem of seeking the approximated operator F_L^{-1} is usually highly-ill-posed (not having a well-defined unique output) and poses great challenges to the reconstruction algorithms" -> what is ill posed? computing an operator approximation or the inverse problem itself? what poses great challenges? the problem or the ill-posedness? and what is meant by output?

- in equation (2), what is the summation over? should the norm be replaced by some pointwise discrepancy? or the sum removed?

- "In this regard, the proposed study provides a positive example to a hopefully definitive answer to this question, which bridges deep learning and conventional tasks in physical sciences." -> I don't undersatnd this sentence

- "the constructional proof of the existence of ILD still relies on the entire NtD mapping Λσ , which again resorts to infiniteness, thus inaccessible in real applications." -> this is just very hard to read and broken

- "If we further assume that there exist a set of feature maps for query, key, and value, e.g., see Choromanski et al. (2021)." -> this is again a broken sentence

- I find the descriptions in 3.3 and 3.4 very difficult to follow. I understand the high-level idea in 3.3 but the intuitive reasoning, the motivation for the different choices, and the interpretation of the harmonic extension would greatly help my understanding.

- I am not listing the numerous misprints and non-idiomatic constructions that further complicate things


### Additional comments and questions

- just before Theorem 2 you state that a CNN cannot generate an output with higher frequencies than the input but this is not true; CNNs such as a U-Net are routinely uses for tasks like image super-resolution. Your statement would be true for pure convolutions but CNNs have nonlinear activations and other architectural details which allow them to generate high frequencies.

- I find it unfair to say that methods like MUSIC or D-bar are ad hoc in the EIT context; one could then similarly state that harmonic extensions as input to a transformer is also ad hoc.

- "However, such a simple closed form of Gθ admitting efficient execution may not be available in practice since some mathematical assumptions and derivation may not hold." which assumptions and derivations are those?

- Could you show the ground truth in Figure 5? (appendix)

- Out of curiosity: is there any relation between harmonic extensions and positional encodings?




**Summary Of The Paper:**

The authors propose to use transformers for electric impedance tomography, with ideas that may apply to a broade range of boundary value inverse problems. EIT is known to be very ill-posed (only log stable) so it is a challenging test for any method. The authors frame the problem as inverting the samples of the NtD map. They draw parallels between integral operators / kernels and the architecture of attention layers. They consider the design of proper spatial inputs to the reconstruction map from the boundary data, and propose harmonic extensions as a solution.


**Summary Of The Review:**

I think this paper contains some nice ideas and I very much appreciate the proper "inverse problems culture" in respecting the structure of the problem. On the other hand, the presentation needs a lot of work, both in the direction of improving the prose and clarity, and in the direction of adapting the style to the ICLR audience, providing the requisite background, and abstracting the most important messages without over-entangling them with the specifics of EIT.

---

> ### Author Response · Authors · 2022-11-18
> **Response to Reviewer 4qwm part 3**
>
> > I find the descriptions in 3.3 and 3.4 very difficult to follow. I understand the high-level idea in 3.3 but the intuitive reasoning, the motivation for the different choices, and the interpretation of the harmonic extension would greatly help my understanding
>
> The original 3.3 and 3.4 are
>
> 3.3 FROM HARMONIC EXTENSION TO TENSOR-TO-TENSOR
>
> 3.4 FROM INDEX MAP INTEGRAL TO TRANSFORMER
>
> These are two main ingredients of our DNN. We explain them one by one.
>
> **Harmonic extension.**
>
> To emphasize the importance of harmonic extension, we now move 3.3 to 3.2 and use Theorem 1 for the motivation.
>
> - Theorem 1 shows the possibility of using harmonic extensions to construct a function approximating the true index function in a pointwise manner. So the information of the piecewise constant function $\sigma$ is hidden inside the harmonic extension of boundary measurements. It gives a hint on the usage of the harmonic functions as the input which results in a tensor2tensor structure.
> - The harmonic mapping can be also thought of as a feature map: lift 1D data to 2D imagins. Figure 1 is  updated to give a more direct visual cue on how harmonic extension works on converting 1D data to a 2D data. A non-rigorous analogy would be the problem of text-to-image, the harmonic extension is like the CLIP+autoencoder in the image generator in Stable Diffusion.
>
> **Integral form of the index function.**
>
> Many concurrent approaches about DL+EIT just add things in the loss instead of taking advantage of the nice design of DNNs invented by the CS community.
>
> - Theorem 1 is theoretically important but does not give practical algorithm. The eigenvalues of the operator $\Lambda_{\sigma} - \Lambda_{\sigma_0}$ are unknown.
> - Direct sample method DSM ((10) in the rebuttal revision) for $L=1$ gives an explicit and computable expression to approximate $\mathcal{I}^D$.  The original DSM is based on mathematical derivation. We agree that intuition of appropriate choice of probing direction $\bf d$ and $\eta_x$ is missing here. These mathematical objects are invented by the inverse problem community, which turn out to yield good reconstruction with some appropriate choices. We can't add more intuitive reasoning either. Sorry about that.
> - What we want to present is that the DSM formula can be recast into the attention operator by introducing three learnable kernels. Then appropriate choices of  $\mathbf{d}$ and $\eta_x$ are leave to the DNN and all learnable. From the implementation point of view, practitioners do not need to know DSM. So in the rebuttal revision, we also move some mathematical formula on  $\bf d$ and $\eta_x$  to the appendix and just emphasize they can be related to $\nabla \phi$ by $Q,K,V$.
>
> Inspired by the reviewer's comment, structure-wise, these two sections mentioned in the question are greatly revamped. They are now Sections 3.2 and 3.4 in the rebuttal revision. Please check some added summary at the end of Sections 3.3 and 3.4 to explain our thread of thoughts. In addition, we have added a more detailed explanation in Appendix D about the original DSM, and how DSM resembles the attention architecture.
>
> > Just before Theorem 2 you state that a CNN cannot generate an output with higher frequencies than the input but this is not true; CNNs such as a U-Net are routinely uses for tasks like image super-resolution. Your statement would be true for pure convolutions but CNNs have nonlinear activations and other architectural details which allow them to generate high frequencies.
>
> We totally agree with the reviewer, we meant actually a single convolution layer in the original statement. Now it has been fixed on Page 7. We have also added an empirical evidence in Figure 9 and Figure 10 to demonstrate how attention-based learners is easier to "bootstrap'' lower frequency features overall than CNN.
>
> > I find it unfair to say that methods like MUSIC or D-bar are ad hoc in the EIT context; one could then similarly state that harmonic extensions as input to a transformer is also ad hoc
>
> We wholeheartedly agree with the reviewer. We originally meant to use "ad hoc" to describe a very specialized method that takes advantage of the mathematical structure (maybe this is a lost in translation?), and we think our proposed method is ad hoc as well. To avoid confusion, we have changed the statement.

---

> ### Author Response · Authors · 2022-11-18
> **Response to Reviewer 4qwm part 2**
>
> > in equation (2), what is the summation over? should the norm be replaced by some pointwise discrepancy? or the sum removed?
>
> Thanks to this question (also by reviewer 5fLN), we have realized that the original expression (2) is confusing. Our original intention was to present a general setup for inverse problems, and it backfires and brought quite a few confusions. In the rebuttal revision, we have rewritten the whole section dedicated to explain this more clearly, just for EIT.
>
> Now, in (2) ((7) in the rebuttal revision), the summation is taken over all available data pairs $(g_l,f_l=\Lambda_{\sigma}g_l)$, $l=1,2,...,L$. The sum is needed since ideally $\sigma$ can be only recovered by infinite boundary data pairs. In addition, The norm can be the $L^2$ norm over $\partial \Omega$. Thanks to the synthetic data setup, these quantities are readily computable using integral quadratures on the mesh (which is essentially pointwise discrepancy weighted by the local mesh size). Based on how the data are given, other choices are possible for other inverse problems.
>
> > "In this regard, the proposed study provides a positive example to a hopefully definitive answer to this question, which bridges deep learning and conventional tasks in physical sciences." I don't understand this sentence.
>
> Thanks for pointing out this clumsy presentation. We originally meant to emphasize that our study is just one example, by no means a definitive answer to this big question: "A natural question to ask is whether and how existing innovative neural architectures, usually designed for other purposes, can be modified conforming with the mathematical nature of the underlying problem, which ultimately leads to structure-conforming DNNs".  We also realize some sentences are too long to convey a clear message. Now in the revision, we change this sentence to:
>
> - *A natural question to ask is how the a priori mathematical knowledge can be exploited to design more physics-compatible DNN architectures. In pursuing the answer to this question, we aim to provide a supportive example that bridges deep learning techniques and classical direct methods improving the reconstruction.*
>
>
> > `the constructional proof of the existence of $\mathcal{I}^D_L$ still relies on the entire NtD mapping $\Lambda_{\sigma}$ , which again resorts to infiniteness, thus inaccessible in real applications.' -> this is just very hard to read and broken
>
> We originally intended to say that an explicit expression for $\mathcal{I}^D_L$ is developed in (46), so it is constructive.
>
> The involved coefficients rely on the spectral information of $\Lambda_{\sigma}$. However, the whole spectrum is nowhere near computable if only finite data pairs are given. We have revised the paragraph below Theorem 1 (Page 5 of the rebuttal revision) to hopefully clarify this issue.
>
> > `` `If we further assume that there exist a set of feature maps for query, key, and value, e.g., see Choromanski et al. (2021).' -> this is again a broken sentence "
>
> We meant to express that these latent representations are assumed to be functions of a positions (feature map with positional embedding as the independent variable), this is a common practice if we want to interpret the attention mechanism mathematically. This has been rewritten on page 7.
>
> > `However, such a simple closed form of $\mathcal{G}_{\theta}$ admitting efficient execution may not be available in practice since some mathematical assumptions and derivation may not hold.' which assumptions and derivations are those?
>
> We have revised the sentence to be more precise. Please refer to the paragraph of Direct Methods on page 3.
>
> >  Could you show the ground truth in Figure 5? (appendix).
>
> Done.
>
> > Out of curiosity: is there any relation between harmonic extensions and positional encodings?
>
> We all agree that this is a sharp comment as we have tested various positional encodings. At the end of the day, the non-learnable Euclidean coordinates work the best (and is used for every model), as it is the underlying grid of the harmonic extensions. We conjecture that positional encodings determine the topology of the latent spaces and contributes to the nice extrapolation capacity of Transformers. Yet we decided to not include these observations in the paper, as it can be a dedicated new study on the role of positional encodings. Some comments on the rationale of the 2D Euclidean grid as PE is added at the end of Section 3.2.

---

> ### Author Response · Authors · 2022-11-19
> **Response to Reviewer 4qwm part 1**
>
> ### Summary of revision
>
> We greatly appreciate reviewer 4qwm for the time and effort reviewing our paper. Below is a summary of the updated parts to address your questions in the rebuttal revision.
>
> > I find the paper quite unclear.
>
> We agree, and we have made a lot of effort to address this in the rebuttal revision. Please check if the new changes make the paper easier to read.
>
> - Inspired by the reviewer's comment, we have completely rewritten the introduction to make the considered operator approximation problem more clear and rigorous. Instead of presenting a general formulation for inverse problems, we have switched to more EIT-dedicated presentation.
>
> - Sections 3.2 and 3.4 have been revised a lot to explain the motivation of using $\nabla\phi$ as the inputs, as well as the rationale behind using DSM to develop attention-like architectures. Due to the page limitation, we have added those details in Appendix D.
>
> - We have added one section (Appendix B) to specifically describe the theoretical background of EIT using an example.
>
> - More details of our network architectures and those to be compared are added in Appendix C.2 on Pages 21 and 22.
>
> - More numerical experiments including comparison with other methods and plots of latent layers (Figures 9 and 10) are added in Appendix.
>
> &nbsp;
> ## Answers to questions
>
> > what is ill posed? computing an operator approximation or the inverse problem itself? what poses great challenges? the problem or the ill-posedness? and what is meant by output?
>
> Here we mean that the inverse operator  $\mathcal{F}^{-1}_L$ is ill-posed. The output refers to the piecewise constant function $\sigma$  reconstructed via  $\mathcal{F}^{-1}_L$.  We believe the confusion might come from our vague notation on spaces $\mathbb{P}$ and $\mathbb M$. So in the revision, we remove them and focus on EIT only. Define the forward operator and inverse operator
>
> $$
> \mathcal{F}: \sigma \mapsto \Lambda_{\sigma}, \quad  \text{ and } \quad  \mathcal{F}^{-1}: \Lambda_{\sigma} \mapsto \sigma,
> $$
>
> where $\sigma$ is a piecewise constant function and $\Lambda_{\sigma}$ is the NtD mapping which can be represented by an infinite dimension matrix $\mathbf{A}_{\sigma}$. Both $\mathcal F$ and $\mathcal F^{-1}$ are well defined.
>
> When there are only finitely many data pairs, the mapping becomes
>
> $$
> \mathcal{F}_L: \sigma \mapsto \\{ (g_1, \Lambda\_{\sigma} g_1), \dots, (g_L, \Lambda\_{\sigma} g_L) \\} \quad \text{ and } \quad \mathcal{F}^{-1}_L: \\{ (g_1, \Lambda\_{\sigma} g_1), \dots, (g_L, \Lambda\_{\sigma} g_L) \\} \mapsto \sigma.
> $$
>
> In view of the matrix representation $\mathbf{A}\_{\sigma}$ of $\Lambda\_{\sigma}$, for $\mathbf{g}_l = \mathbf{e}_l$, $l=1,...,L$, with $\mathbf{e}_l$ being unit vectors of a chosen basis, $(\mathbf{f}_1,\dots,\mathbf{f}_L)$ only gives the first $L$ columns of $\mathbf{A}\_{\sigma}$.  It is possible that two matrices $\mathbf{A}\_{\sigma}$ and $\mathbf{A}\_{\tilde\sigma}$ have similar first $L$ columns but $\\|\sigma - \tilde \sigma\\|$ is large. That is, the same boundary measurement may correspond to different $\sigma$, or a small perturbation in the measurement may result in large variance in $\sigma$. So the inverse operator $\mathcal{F}^{-1}_L$ is ill-posed.  We have also added a concrete example in Appendix B to illustrate the theoretical background/difficulty of EIT.
>
> Approximating an ill-posed operator is challenging, and if this operator is not even well-defined, trying this approximation problem can be a misstep into a dangerous territory. So we remove some misleading sentences on approximating  $\mathcal{F}^{-1}_L$.
>
> However, instead of approximating the ill-posed $\mathcal{F}^{-1}\_L$, the proposed operator learning approach aims to "learn" the behavior of the operator on a compactly sampled data set $\mathbb{D}:=\\{ \sigma^{(k)}\\}_{k=1}^N$. Then the problem becomes to approximate
>
> $$
> \mathcal{F}^{-1}\_{L,\mathbb{D}}: \\{ (g_1, \Lambda\_{\sigma^{(k)}} g_1), ..., (g_L, \Lambda\_{\sigma^{(k)}} g_L) \\} \mapsto \sigma^{(k)}, \quad k= 1,\dots,N.
> $$
>
> In this way, the ill-posed problem is tamed to be become a (relatively) well-defined data interpolation problem. The incomplete information of $\Lambda_{\sigma}$ due to a small $L$ for one single $\sigma$ is compensated by a large $N\gg 1$ sampling of different $\sigma$'s.
>
> For the forward PDEs, this is also explained in [1] (around equation (2.1)), from which the best forward PDE operator learner Fourier Neural Operator takes the data generation.
>
> [1]: N. H. Nelsen, A. M. Stuart; The Random Feature Model for Input-Output Maps between Banach Spaces, SIAM J. Sci. Comput., 43(5), A3212–A3243.

---

> > ### Comment · Reviewer_4qwm · 2022-11-22
> > **thank you for the thorough revision**
> >
> > I thank the authors for earnestly engaging with my critical remarks. I think that this substantially revised manuscript is a great improvement over the original and I am raising my score accordingly.
> >
> > (I would still recommend to carefully polish the prose.)

---

### Author Response · Authors · 2022-11-17
**Overall comments on the rebuttal revision**

We appreciate reviewers 4qwm, 5fLN, CBcr, and ne3Z for the effort and time spent in reviewing our paper and the suggestions helping us to improve the paper. After carefully stuyding the reviewers' comments, we realized that our original presentation was too ambitious trying to be general. The original introduction tries to lay out the blueprint using a general boundary value inverse problem, yet it backfires a lot and caused much confusion. We also realized that due to extra paragraph overhead caused by two sets of notations in the original submission, we have to omit some details and put those in the Appendix, thus not very friendly to the audience from the greater DL+IP community.

- Thanks for the suggestions by 4qwm, 5fLN, and CBcr, the introduction is now completely overhauled: the presentation of the inverse problem focuses on EIT only, an updated Figure 1 is added to illustrate the overall pipeline.
- Sections 3.2 and 3.4 have been revised a lot to explain the motivation of using the PDE-based feature map $\nabla\phi$ as the inputs, as well as the rationale behind using DSM to develop attention-like architectures. Due to the page limitation, we have added a more detailed explanation in Appendix D.
- We hope that the rebuttal revision lays out our thread of thoughts more clearly:
    - boundary value inverse problem is hard (the operator itself is ill-posed) &rArr;
    - so we use operator learning to tame the ill-posedness &rArr;
    - but how to find suitable data format? (harmonic extension fits the tensor2tensor pipeline) &rArr;
    - how about the basic block in the network? (kernel integration as attention) &rArr;
    - why it may be good for this problem? (DSM and attention have similar forms and have global dependence).
- Thanks to 5fLN and CBcr, we have added the benchmark results for two neural operators featured in ICLR 2022: adaptive FNO2d with several token mixing layers in frequency, and MultiWavelet Neural Operator.
- Plots of latent representations (Figures 9 and 10) are added in the Appendix to show empirically how Theorem 2 works in action.
- Thanks to 5fLN, we have added Appendix B to present a more detailed review of EIT through an example to help understand the difficulty of this problem.
- Thanks to CBcr and ne3Z, we have updated much more details about the training and network architecture comparisons in Appendix C.
- We greatly appreciate ne3Z for reviewing our paper again, and we think for this time we have a more satisfactory answer to the questions about synthetic data.
- Source code update in the supplemental material: memory and FLOPs profiling, AFNO2d, MWO, DeepONet2d, a test for mesh-invariance.
- Many broken sentences and typo have been fixed, thanks to reviewers' reading.

For detailed responses to each question from each reviewer please see below.

---

### Decision · Program_Chairs · 2023-01-20

**Decision:**

Accept: poster

**Justification For Why Not Higher Score:**

This work only study a very specific class of boundary value inverse problem and requires several assumptions. The quality of the paper could be further improved by investigating more general cases.

**Justification For Why Not Lower Score:**

The paper is well-motivated. The method is solid and interesting. During the rebuttal, they addressed most reviewers' concerns and largely improve the paper.

**Metareview: Summary, Strengths And Weaknesses:**

This paper presents a new method for solving a class of boundary value inverse problems, electrical impedance tomography (EIT), based on Transformer. This paper makes contributions in both theoretical and experimental sides. Theoretically, the authors prove that a modified attention module can represent target functions with higher frequencies than the input, and show the connections between the index function integral and the attention mechanism in Transformer. Experimentally, the authors obtain the strong empirical results on EIT benchmark tests with the proposed U-Integral Transformer architecture under various settings.

The reviewers agree that this work is novel and interesting (4qwm, 5fLN, CBcr, ne3Z), and the strong empirical performances look promising (4qwm, 5fLN, CBcr). Some concerns are raised, particularly on the presentation quality of the paper (4qwm, 5fLN, CBcr), and most reviewers feel the rebuttal revision addresses the concerns. In conclusion, we recommend acceptance.

**Note From Pc:**

if the above contains the word "oral" or "spotlight" please see: "oral" presentation means -> notable-top-5% and "spotlight" means -> notable-top-25%. As stated in our emails, we are disassociating presentation type from AC recommendations